Manuscript prepared for Earth Syst. Dynam.
with version 2014/09/16 7.15 Copernicus papers of the LATEX class copernicus.cls.
Date: 31 August 2017

# More Homogeneous Wind Conditions Under Strong Climate Change Decrease the Potential for Inter-State Balancing of Electricity in Europe

Jan Wohland[1,2], Mark Reyers[3], Juliane Weber[1,2], and Dirk Witthaut[1,2]

[1]Forschungszentrum Jülich, Institute for Energy and Climate Research (IEK-STE), 52428 Jülich, Germany
[2]Institute for Theoretical Physics, University of Cologne, 50937 Cologne, Germany
[3]Institute for Geophysics and Meteorology, University of Cologne, Germany

*Correspondence to:* Jan Wohland (j.wohland@fz-juelich.de)

**Abstract.** Limiting anthropogenic climate change requires the fast decarbonisation of the electricity system. Renewable electricity generation is determined by the weather and is hence subject to climate change. We simulate the operation of a coarse-scale fully-renewable European electricity system based on downscaled high resolution climate data from EURO-CORDEX. Following a high emission pathway (RCP8.5), we find a robust but modest increase (up to 7%) of backup needs energy in Europe until the end of the 21st century. The absolute increase of the backup needs energy is almost independent of potential grid expansion, leading to the paradoxical effect that relative impacts of climate change increase in a highly interconnected European system. The increase is rooted in more homogeneous wind conditions over Europe resulting in extensive parallel intensified simultaneous generation shortfalls. Individual country contributions to European generation shortfall increase by up to 9 TWh/y, reflecting an increase of up to 4%. Our results are strengthened by comparison with a large CMIP5 ensemble using an approach based on Circulation Weather Types.

## 1 Introduction

Massive reductions of greenhouse gas emissions are needed in order to reach the temperature goals
defined in the Paris Agreement (UNFCCC, 2015; Schleussner et al., 2016b). With a share of around
35% of current emissions being caused by the electricity system (Bruckner et al., 2014), its decarbonisation is the key to any mitigation strategy. However, today's pledges are not yet sufficient to
limit warming to below 2°C, not to mention 1.5°C (Rogelj et al., 2016).

In addition to the need of mitigating carbon emissions, a second interaction between the energy
system and the climate system exists and becomes increasingly important with higher penetrations
of renewable energies. Volatile renewable energy generation is driven by weather conditions which
are subject to climate change. Large backup facilities are needed to guarantee a stable supply of
electricity during periods of low wind and solar power generation (Rodriguez et al., 2014). Furthermore, climate change affects the demand for electric power (Auffhammer et al., 2017) as well as the
operation conditions for thermo- and hydroelectric power plants which serve as backup (van Vliet
et al., 2016, 2012). However, feedback effects of large-scale wind fleets on atmospheric flows are
limited (Vautard et al., 2014).

In line with the Paris Agreement, the scientific community is increasingly interested in differentiating climate impacts at 1.5°C and 2°C (Schleussner et al., 2016a; James et al., 2017) and the IPCC
currently prepares a special report on 1.5°C. However, many low-carbon pathways rely on negative
emissions during the second half of this century (Rogelj et al., 2015; van Vuuren et al., 2016), although their feasibility at scale remains debated (Anderson and Peters, 2016). Future emissions from
existing $CO_2$-emitting infrastructure (Davis et al., 2010) and current political developments in the
US (Trump, 2017), among others, might impede fast decarbonisation. Different climatic futures are
hence plausible and mitigation strategies need to work in all of them. Therefore, we are led to the
question: How sensitive is a fully-renewable electric power system to climate change? In particular:
How severely could strong climate change impact such a system?

Anthropogenic climate change affects the large-scale atmospheric flow and thus the operation conditions for renewable power generation. State-of-the-art global climate models reveal that changes
in zonal wind depend on the temperature structure of the lower atmosphere (Haarsma et al., 2013)
and that zonal-mean zonal wind and eddy kinetic energy decline almost linearly in time due to polar
amplification (Coumou et al., 2015). There are also natural sources of variability on up to decadal
timescales. Some of them originate from ocean-atmosphere interactions in the Atlantic and are potentially predictable (Haekkinen et al., 2011; Peings and Magnusdottir, 2014). The North Atlantic
Oscillation has been shown to directly influence the operation of inter-connected renewable electricity systems (Ely et al., 2013). Predictability of such natural variations is of great interest for system
integration and efforts are undertaken to assess and improve forecasting skills (Moemken et al.,
2016).

To assess the impact of climate change on the operation of renewable power systems, downscaled climate model output is needed. It comes at a high temporal and spatial resolution and is better suited than global model output to capture local features such as land-sea transitions or mountains (Rummukainen, 2016). Temporal resolutions at the sub-daily scale are needed since electricity consumption varies strongly during the day. Changes in wind energy yields and capacity factors have been assessed based on dynamical (Tobin et al., 2015, 2016) and statistical-dynamical downscaling outputs (Reyers et al., 2015, 2016). Tobin et al. (2016) evaluate the EURO-CORDEX data archive and find that changes in the annual wind energy yield across Europe are at the order of 5% and models do not agree on the sign of change. Following a different approach that allows for the inclusion of the output of 22 global climate models, Reyers et al. (2016) report an increasing intra-annual gradient between winter and summer wind generation and different trends in Northern and Central Europe as compared to Southern Europe.

Assessing changes in solar power generation is arguably more difficult due to, among others, unresolved processes in relatively coarse climate models and uncertain parameterizations (e.g. (Chiacchio et al., 2015; Herwehe et al., 2014)) (e.g., Chiacchio et al., 2015; Herwehe et al., 2014). Acknowledging this difficulty and associated uncertainties, an evaluation of the EURO-CORDEX data finds limited impacts of climate change on solar photovoltaics (PV) potentials (Jerez et al., 2015). Southern Europe, having the highest potential for PV, sees only small changes, as an increase in downwelling irradiation is counteracted by a decreasing efficiency due to warming. In contrast, the output of concentrating solar power systems (CSP) is expected to increase by around 10% because the efficiency of CSP increases with temperature (Crook et al., 2011).

While wind and solar power sources have shown a remarkable development in the last decades, system integration remains a huge challenge (Huber et al., 2014). In a highly renewable power system the timing of generation events becomes crucial for the system. Even in an European electricity system that is on average fed by 100% renewables, roughly one quarter of the energy is produced at the wrong time and has to be curtailed (Rodriguez et al., 2014, 2015a).

It is thus necessary to consider indicators such as the variability and synchronicity of generation in addition to total energy yields (Monforti et al., 2016; Bruckner et al., 2014; Bloomfield et al., 2016). Several validated timeseries of renewable generation based on reanalysis data are available to assess the power system operation (Pfenninger and Staffell, 2016; Staffell and Pfenninger, 2016; Gonzalez Aparcio et al., 2016). However, these data sets are restricted to current climatic conditions and might thus be misleading for long-term planning of the electricity system.

In this article we study the impact of climate change on the operation conditions for future fully-renewable power systems. We combine the analysis and simulation of power systems with high-resolution regional climate modeling results to quantify changes in wind power generation. We adopt a coarse scale view on the power system to uncover the large-scale impacts of climate change. The coarse scale perspective neglects details that are irrelevant for the balancing of demand with wind

generation such as supply of apparent power or different voltage levels in the grid. The focus of this study is to In particular, we address the potential of trans-national power transmission to cover regional balancing needs.

Our results reveal the sensitivity of fully-renewable power systems to climate change. They should not be mistaken with a forecast and rather be considered a thought experiment to assess potential risks and to answer the question: What happens to a fully-renewable electricity system if mitigation actions are ineffective or come too late?

## 2 Methods

**Modeling the operation of a fully-renewable power system under climate change**

The power generated by wind turbines and solar photovoltaics is determined by the weather such that its variability crucially depends on atmospheric conditions (see, e.g. Bloomfield et al. (2016)). How does climate change affect these conditions and the challenges of system integration?

To assess the impact of strong climate change, we simulate the operation of a fully-renewable power system making use of high resolution climate projections. We use the EURO-CORDEX en-
100 semble containing output of Global Circulation Models (GCMs) which has been dynamically down-scaled to a finer resolution (Jacob et al., 2014) to quantify changes in wind power generation. The ensemble contains five GCMs (HadGEM2-ES, CNRM-CM5, EC-EARTH, CM5A-MR amd MPI-ESM-LR) which are all downscaled by the regional climate model RCA4 (Strandberg et al., 2015). The GCM output is part of the Climate Model Intercomparison Project Phase 5 (CMIP5) and pub-
105 licly available (Taylor et al., 2011). We use near-surface wind speeds at $0.11°$ spatial and 3 h tempo-ral resolution and hence capture intra-day effects. In the spirit of a sensitivity analyses analysis, we evaluate the representative concentration pathway RCP8.5. It describes atmospheric greenhouse gas concentrations following a business-as-usual strategy and leads to approximately $4.3°C$ warming at the end of the century as compared to pre-industrial values (Stocker et al., 2013). In view of inter-
110 model spread and other uncertainties, a strong climate change scenario bears the advantage of high signal-to-noise ratios.

The approach used in this study is illustrated in Fig. 1. The climate data is used to calculate the aggregated wind power generation time series for each country in the interconnected European power grid (grey circles in Fig. 1a). Near-surface wind speeds are scaled up to hub height (80 m)
based on a power law and a standard power curve is used to obtain the power generation of the wind turbines, both as in Tobin et al. (2016) (see also Supplementary Material A). The power curve assumes a cut-in velocity of $3.5 \ \mathrm{m/s}$, a rated velocity of $12 \ \mathrm{m/s}$ and a cut-out velocity of $25 \ \mathrm{m/s}$. Wake losses are not accounted for. The country-wise aggregated wind power is obtained by summing the generation of 100 MW wind parks until the system is fully-renewable on average. The wind park
size was chosen as a compromise between increasing turbine capacities (Wiser et al., 2016) and the need for a sufficient amount of distinct parks. Wind parks are deployed semi-randomly following the approach of Monforti et al. (2016). In order to single out climate change induced alterations, we fix the technological parameters such as hub heights or turbine efficiencies, and we do not account for changes in the consumption such as load shifting or sector coupling throughout the 21st century.
Tests including validated historical PV timeseries (Pfenninger and Staffell, 2016) reveal that the inclusion of PV does not change the overall results (see Supplement B). For the sake of simplicity, we thus decide to restrict the analysis to wind-driven power systems in this paper.

Wind power generation is strongly fluctuating on various time scales as shown in Fig. 1c. In periods of scarcity, energy has to be imported from other countries or generated from local dispatchable power plants. We refer to the latter as backup energy. In the situation depicted in Fig. 1a, scarcity in Southern Europe can mainly be compensated by imports from Northern Europe. Trans-national balancing of this kind often requires large transmission capacities. Moreover, the import of electric energy requires a respective exporter which has a surplus at the same time. Backup needs energy in future renewable power systems are is thus essentially determined by the temporal and spatial heterogeneity of wind and solar power throughout the system.

In addition to enhanced spatial balancing via im- and exports, an extension of storage facilities will reduce backup needs energy (Rasmussen et al., 2012). But storage assets are more costly than grid expansion (Schlachtberger et al., 2017; Brown et al., 2016). Since a cost-optimal solution will thus favor grid expansion, we focus on spatial effects and trans-national balancing. An assessment of climate change effects on storage following a similar approach is presented by Weber et al. (2017).

To quantify backup needs energy, we adopt a coarse-scale view of the transmission system (see, e.g. Rodriguez et al. (2015a, 2014)). We consider each country $i$ to be a node in the European transmission network and define a nodal mismatch for each point in time $t = 1, 2, \ldots$ as

$$M_i(t) = P_i(t) - D_i(t), \tag{1}$$

where $P_i(t)$ is intermittent renewable generation and $D_i(t)$ is the load (here: hourly data for 2015 from European Network of Transmission System Operators for Electricity (ENTSO-E) averaged over 3h timesteps (European Network of Transmission System Operators for Electricity, 2015)). The assumption of a fully-renewable system means that all countries generate as much electricity as needed on average ($\int_{t_s}^{t_e} M_i(t)dt = 0$). Furthermore, we We assume all countries to run a loss-free and unlimited transmission network within their boarders.

If a country has a negative mismatch ($M_i < 0$, red circles in Fig. 1d), it tries to import energy. If it has a positive mismatch ($M_i > 0$, green circles in Fig. 1d), it tries to export energy. For each country $i$ the power balance must be satisfied:

$$M_i(t) + B_i(t) + F_i(t) = C_i(t), \tag{2}$$

The mismatch $M_i$ can be compensated either by power generation from conventional backup power plants ($B_i \geq 0$), the curtailment of renewable power generation ($C_i \geq 0$) or by imports ($F_i > 0$) or exports ($F_i < 0$). To utilize renewable generation in an optimal way, countries will first try to balance power using im- and exports. However, a perfect balancing of all nodes is impossible if there is a continent-wise shortage or overproduction. Furthermore, cross-boarder flows along lines are bound by the directional Net Transfer Capacities (NTCs; see Supplement A for details), which

may also impede balancing for some nodes. Power balance must then be satisfied by local means: In the case of a shortage, power must be backed up by conventional generators ($B_i > 0$). where $F_i$ represents imports ($F_i > 0$) or exports ($F_i < 0$) to/from country $i$. Cross-boarder flows along lines are bound by the directional Net Transfer Capacities (NTCs; see Supplement A for details). If overall shortage or line limits prohibit sufficient imports, power can also be backed up locally ($B_i \geq 0$). Similarly, if excess power can not be exported, it has to be curtailed ($C_i \geq 0$). We recognize that the technical details of backup generation often matter for implementation (Schlachtberger et al., 2016) but we focus on gross electricity needs in this study.

For each time step we determine the system operation which minimizes backup power and thus macroeconomic costs as well as greenhouse gas emissions. To assess the impact of climate change, we compare future backup energy needs to historical values. Time frames of 20 year duration are chosen to account for natural climatic variability (see Table 1). Time frames are chosen to contain 20 years in order to capture natural variability of the climate system on a multi-year timescale while still ensuring that elapsed time between periods is long enough to consider them distinctly (see Table 1). Since GCMs do not reproduce natural variations synchronously (Farneti, 2017), robust signals found in the ensemble are very unlikely to be rooted in natural variations with a recurrence time of a couple of decades (such as the Atlantic Meridional Oscillation or the North Atlantic Oscillation; see Peings and Magnusdottir (2014) for a discussion of their role in mediating atmospheric conditions). The backup energy $E_\text{B}$ per period is defined as the sum over all backup powers in a given period:

$$E_\text{B}(\text{period}) = \sum_{t \in \text{period}} \min \sum_i B_i(t), \tag{3}$$

such that Eq. (2) is satisfied for all countries $i$ and the line limits are respected.

The European amount of backup energy is identical to the amount of curtailment over a full period. This is a direct consequence of the assumptions made and can be formally derived by summing Eq. 2 over all countries and integrating over an entire period. Since $\int_{t_s}^{t_e} M_i(t)dt = 0$ (each country is fully renewable on average) and $\sum_i F_i = 0$ (all imports to one country $F_j = c$ are exports from another $F_k = -c$) it follows that:

$$\int_{t_s}^{t_e} \sum_i B_i(t)dt = \int_{t_s}^{t_e} \sum_i C_i(t). \tag{4}$$

A change of the backup energy thus directly implies a change in total curtailment.

We use climate model ensembles to account for model uncertainties. Interpretating the ensemble output by means of the ensemble-mean can be misleading as a single model might dominate the ensemble. In such cases, the model-mean would be in disarray with the majority of models and hence would not be representative for the ensemble. We thus assess the robustness of changes by means of inter-model agreement. We label a signal *robust* if all models agree on the sign of change

and use *high agreement* if all but one model agree. In the evaluation of the large CMIP5 ensemble
we adopt language defined for the latest IPCC report and label a change *likely* if at least 66% of
models agree (Mastrandrea et al., 2010).

A variety of studies have analyzed transmission and backup needs energy in future renewable
power systems and cost-optimal transition pathways in a similar way (Rodriguez et al., 2015a, 2014,
2015b; Becker et al., 2014; Rasmussen et al., 2012; Schlachtberger et al., 2016; Hagspiel et al.,
2014). But the potentially crucial role of changes in climatic conditions have not yet been assessed
in this context. The remainder of this article focuses on the quantification of impacts to the power
system, a correlation analysis of the wind resource and an assessment of the larger CMIP5 ensemble
to contextualize our findings.

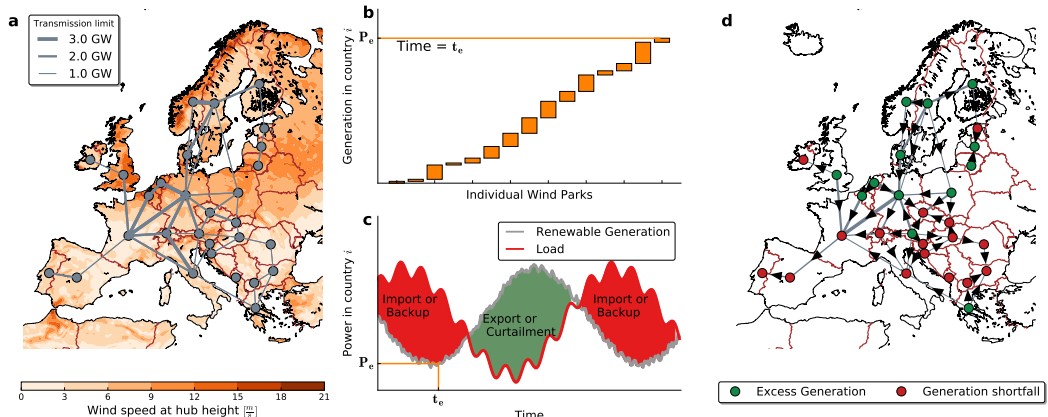

**Figure 1. Approach of the study.**
**a)** Wind fields from high-resolution climate models and the 2010/2011 Net Transfer Capacities are used as
input to the model. **b)** The wind speeds are first translated into generation of individual wind parks using local
wind fields . In a second step the generation is and then aggregated to a national level for each country. **c)** In
combination with country-specific load data, the nodal mismatch for every country and timestep is computed.
If generation exceeds the load (green area), countries can export energy until lines reach their transmission ca-
pacity. Remaining energy has to be curtailed (dumped). If generation is lower than load, electricity is imported.
If importing is not an option due to transmission limits or lack of available excess energy in other countries,
backup energy has to be provided by dispatchable power plants (e.g. gas turbines or other thermoelectric plants).
**d)** We assume a controllable European power transmission grid. A minimization of the total backup energy of
all countries then yields a flow pattern in Europe. In the shown case, strong winds over the North Sea lead to
high generation in this region while there is little generation in the southern part of Europe. Energy is hence
mainly transported from the North Sea region to Southern Europe and the high transmission needs lead to an
operation of almost all lines at their maximum.

**Table 1.** Periods are chosen to contain 20 years in order to capture natural variability of the climate system on a multi-year timescale while still ensuring that elapsed time between periods is long enough to consider them distinctly. Since GCMs do not reproduce natural variations synchronously (Farneti, 2017), robust signals found in the ensemble are very unlikely to be rooted in natural variations with a recurrence time of a couple of decades (such as the Atlantic Meridional Oscillation or the North Atlantic Oscillation; see Peings and Magnusdottir (2014) for a discussion of their role in mediating atmospheric conditions). Periods used in this study. The reference period ref ends before 2005 because GCMs in CMIP5 are driven by historic emissions only until this date and follow different representative concentration scenarios afterwards.

| Period name | $t_{start}$ | $t_{end}$ |
|---|---|---|
| ref | 1985 | 2004 |
| midc | 2040 | 2059 |
| endc | 2080 | 2099 |

## 3 Results and Discussion

### 3.1 Energy: Increasing backup needs energy

A cost-efficient way of power balancing is given by trans-national im- and exports. Remarkably, we find that strong climate change impedes the potential of this balancing measure in most of Europe (cf. Fig. 2). We report that backup needs energy in Europe increases under strong climate change by the end of the century. This finding is robust across all EURO-CORDEX ensemble members. The increase implies more excess energy and also more curtailment since we consider a scenario where 100% of electricity is generated from renewables on average. Since we consider a scenario where 100% of electricity is generated from renewables on average, an increase of backup energy is accompanied by an increase of excess energy which has to be curtailed.

To uncover this effect we simulate backup needs energy for different scenarios of the development of the transnational grid quantified by the Net Transfer Capacities (NTCs). We allow for a homogeneous scaling of transmission capacity by multiplying NTCs with a factor $\alpha$. Without any grid, approximately 45 % of the wind-energy is produced at the wrong time and thus has to be curtailed and backed up later on. A strong grid extension ($\alpha \gg 1$) clearly reduces total balancing needs backup energy to about 27% (cf. Fig. 2a). However, all models report an increase of backup energy at the end of the century. The the effect of climate change is almost independent of a grid extension: The absolute increase of backup energy until end of century is largely independent of the expansion coefficient $\alpha$ for three out of five models (cf. Fig. Fig. 2b). Hence, the relative increase of backup needs energy paradoxically becomes even more pronounced for a strongly interconnected Europe (cf. Fig. Fig. 2c). Highly connected systems can suffer from an increase of backup needs energy of up to 7%. There is considerable inter-model spread regarding the magnitude of change which varies by up to one order of magnitude depending on the climate model (see Fig. 2b, $\alpha = \infty$). In particular, changes for CNRM are generally weak and HadGEM2 features only a slight overall increase with grid expansion. But remarkably, all models agree on the sign of change at the end of the century such that we consider the direction of change very likely.

In conclusion, we find that the effectiveness of transnational balancing decreases due to climate change. This decrease is due to more homogeneous wind generation as we will show in the climate section of this paper. Moreover, a control simulation including PV generation from Pfenninger and Staffell (2016) yields similar results although the magnitude of change is reduced by roughly a factor of 2 and only 4 out of 5 models agree (cf. Supplementary B). Results are barely sensitive to changes in the load timeseries as an assessment using constant loads reveils reveals (cf. Supplementary C).

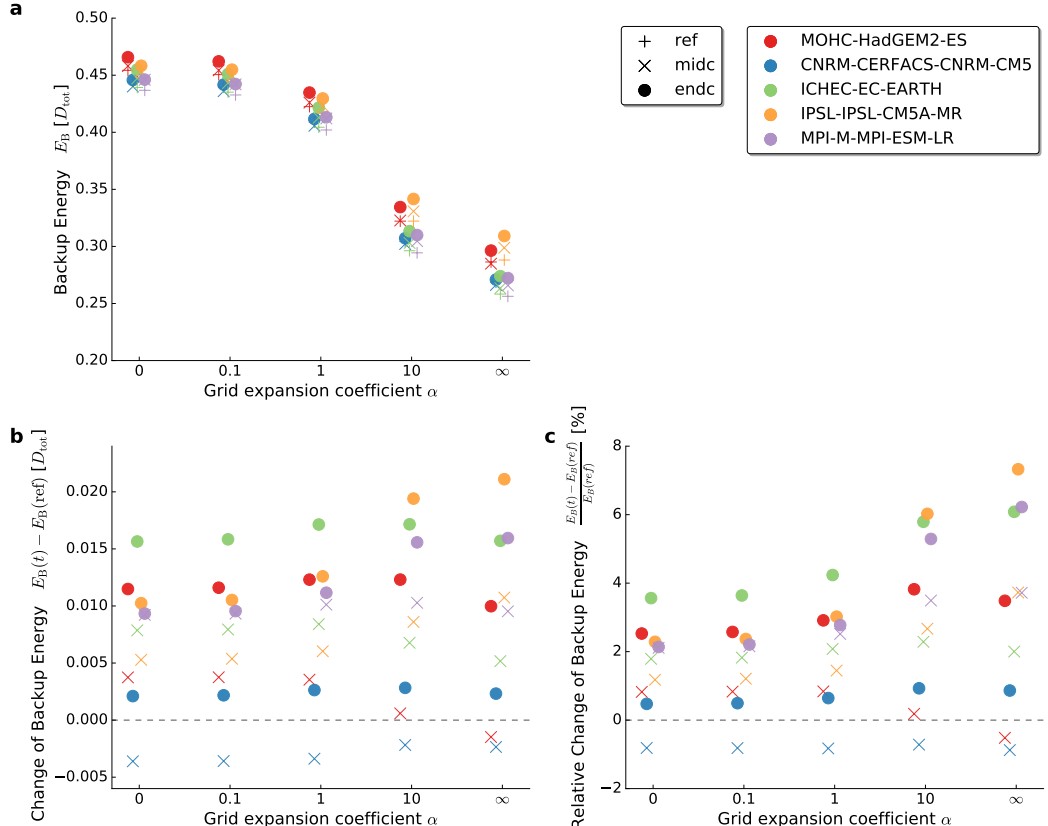

**Figure 2. The impact of climate change on backup energy under different grid expansion scenarios.**
Different realizations of the European inter-state grid expansion are given by the grid expansion coefficient
$\alpha$. While $\alpha = 0$ denotes the isolated case without inter-country transmission network, $\alpha = 1$ reproduces the
configuration as of today and $\alpha = \infty$ represents unlimited European transmission. Different markers refer to
distinct 20 year time periods (see Table 1), colors denote different climate models. **a)** Backup energy as a func-
tion of grid expansion expressed in units of the total European load $D_{\text{tot}} = \int \sum_i D_i(t)dt$. **b)** Absolute change
of backup energy by the end of the century. **c)** Relative change of backup energy by the end of the century.

**a)** Backup energy decreases monotonously with grid expansion. Without any grid, approximately 45 % of the
wind-energy is produced at the wrong time and thus has to be curtailed and backed up lateron. This number can
be theoretically reduced to roughly 27% by grid extension.

**b)** All models report an increase of backup energy at the end of the century. This increase is approximately in-
dependent of the grid expansion for 3/5 models. For the other two models the increase is even more pronounced
for a strongly interconnected grid (large $\alpha$).

**c)** The relative change of backup energy features a steeper increase with grid expansion as compared to **b**.
Highly connected systems can suffer from an increase of backup needs energy of up to 7%.

### 3.1.1 Spatial distribution of Mismatch contributions

To obtain a more detailed view, we evaluate trans-national balancing potentials separately for each country. We calculate the likeliness that a given country has a local scarcity ($M_i < 0$) while Europe as a whole suffers from a lack of generation ($\sum_i M_i < 0$). This corresponds to events where a country would favor importing electricity but can not due to a continent-wide scarcity. These events require conventional backup even in the case of unlimited transmission infrastructures and thus give a lower bound for backup needs energy. The approach allows us to identify those countries which are most responsible for overall scarcity. Mathematically speaking, we restrict our analysis to timesteps $T_i$ with local and Europe-wide scarcity:

$$T_i = \left\{ t : \left( \sum_j M_j(t) < 0 \text{ and } M_i(t) < 0 \right) \right\}. \tag{5}$$

The negative mismatch contribution occurence $\nu_i$ corresponds to the joint probability of country $i$ and Europe experiencing generation shortfall at the same time:

$$\nu_i = \frac{\sum_{t \in T_i}}{N_T}, \tag{6}$$

where $N_T$ is the number of timesteps. We define the annual energy that is lacking (i.e., generation shortfall) in country $i$ during European scarcity as

$$L_i = \frac{\sum_{t \in T_i} |M_i(t)|}{20y}, \tag{7}$$

where we chose the absolute value of $M_i$ for convenience of interpretation. $L_i$ is given in TWh/y A high value of $L_i$ characterizes a country which would favor to import a lot of energy during European scarcity whereas a low value of $L_i$ indicates a country whose generation shortfall can often be balanced by imports. In order to compare values of $L_i$ with loads, we provide country values for $D_i$ in the Supplementary Material E. The European sum is $\sum_i D_i \approx 3100$ TWh.

Values for $\nu$ and $L$ during the reference period are shown in Fig. 3a,b. Large consumers like Germany and France are also the dominant contributors to European scarcity in terms of missing energy (cf. Fig. 3a). The German contribution corresponds to approximately 8% of the European annual load of 3100 TWh. However, the role of these countries, for example, in comparison to Eastern Europe or Benelux, is less pronounced if only the occurrence of negative mismatch events $\nu$ is considered (cf. Fig. 3b). The reason for their strong impact on $L$ is thus primarily rooted in the high absolute values of their mismatches rather than their frequency. Moreover, a large consumer also has a bigger influence on the Europe-wide mismatches which implies that the conditions in Eq. (5) are not independent. For example, the European mismatch can be negative because of an elevated

German mismatch and in such a situation a high contribution to $L$ would be observed. Interestingly, there is considerable spread regarding $\nu$ in different countries (Fig. 3b). Greece and Norway contribute the least often to European scarcity (less than 40%) while Central Europe contributes around 50 - 60 % of the time.

Next, we focus on changes until the end of the 21st century

$$\Delta\nu_i = \nu_i|_{\mathrm{endc}} - \nu_i|_{\mathrm{ref}} \text{ and } \Delta L_i = L_i|_{\mathrm{endc}} - L_i|_{\mathrm{ref}}. \tag{8}$$

In France, Benelux, Scandinavia, the British Isles and most countries in Central Europe the negative mismatch contribution occurence $\nu$ and the respective negative energy contribution $L$ increase (cf. Fig. 3c,d). In these countries it becomes more likely that a Europe-wide scarcity coincides with a local scarcity and the amount of required backup energy increases. In turn, these countries can not alleviate the overall shortage by exporting excess generation. This points to a stronger homogeneity of wind power generation in Central Europe which is discussed in more detail below. An increase of the occurence $\nu$ can also be observed for Eastern and South Eastern Europe, excluding Greece, with high inter-model agreement (cf. Fig. 3d). However, these increases are weak in terms of energy contributions (cf. Fig. 3c).

An opposite trend is observed in Spain, where transnational balancing is facilitated as negative mismatch contributions $L$ become weaker (cf. Fig. 3c). At the same time, models generally disagree on the sign of change regarding $\Delta\nu$ (cf. Fig. 3d). Combined, this indicates weaker but not less frequent negative contributions of Spain. Moreover, Greece shows favourable changes for the European system in terms of energy contributions and occurences with a high inter-model agreement (cf. Fig. 3c,d). This finding is particularly interesting as Grams et al. (2017) show that a combination of wind parks allocated in the North Sea and the Balkans allows to reduce volatility substantially under current climatic conditions. Based on our results, this positive effect from incorporating the Balkans would further be enhanced under strong climate change.

We stress that our findings do not refute the efficiency of transmission grid expansions in general. In any case backup needs energy decreases monotonously with the grid expansion, but the magnitude of the decrease is subject to climatic conditions. Furthermore, we assume a homogeneous expansion of the grid, although an optimal system design will probably lead to heterogeneous grid expansions and heterogeneous allocations of generation capacities. Our results suggest that such an optimal system will include stronger interconnections to Spain and Greece to reduce backup needs energy . Also on a country level, certain extensions can be incentivized while others are downgraded. For instance, for France it can become more favorable to extend the connections to Spain rather than to Germany (cf. Fig 3c). Despite that and in light of regulatory and strong social acceptance issues regarding grid extensions (Battaglini et al., 2012), we consider a future grid which resembles the current one in its fundamental characteristics a reasonable first guess.

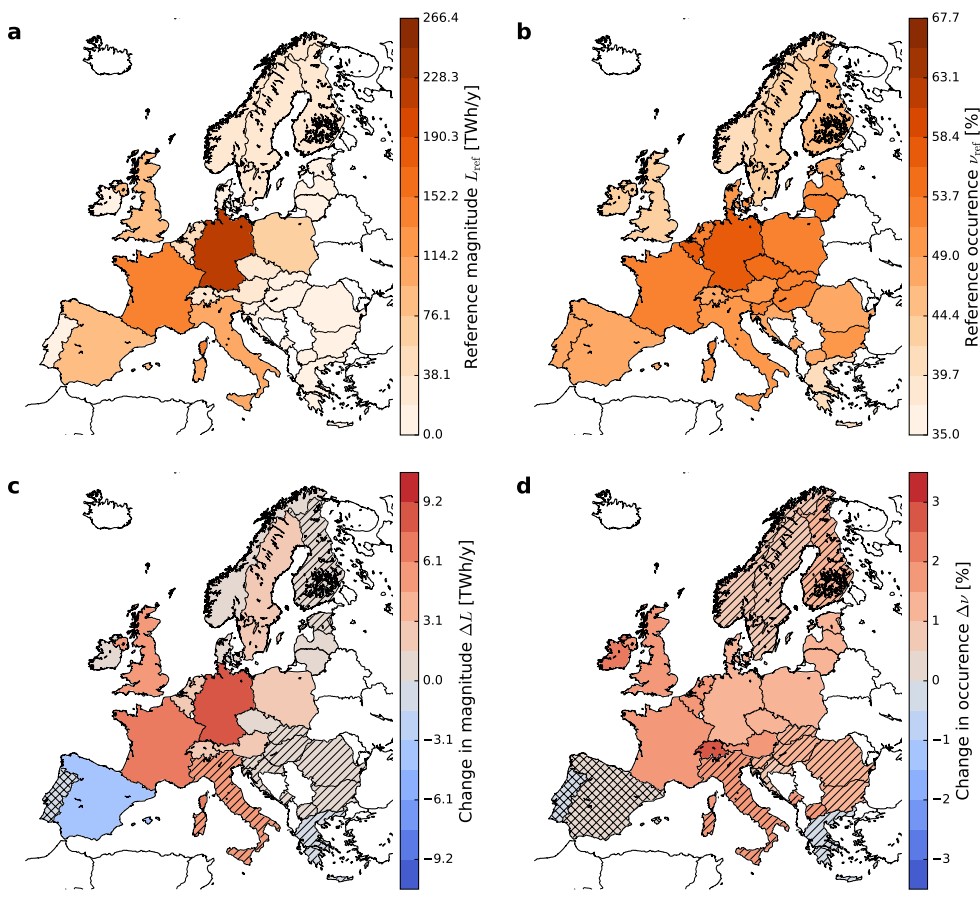

**Figure 3. Country contributions in times of overall and local generation shortfall and their change until end of century.** Values denote the inter-model mean. Hatches indicate inter-model agreement as follows: no hatches indicate perfect agreement on sign of change, striped: 4 out of 5 models agree, crosses: less than 4 agree. **a)** Lacking Energy $L_{ref}$ during local and overall scarcity in the reference period (see Eg. (7)). **b)** Simultaneous occurence of local and overall generation shortfall $\nu_{ref}$ (see Eq. 6). **c)** & **d)** show changes of the quantities given in **a)** & **b)** until end of century (see Eqs. 8). Red colours denote unfavourable changes (stronger or more frequent contribution of a country to overall scarcity) while blue colors denote favourable changes.

## 3.2 Climate: Increasing correlations of the wind resource

As reported above, we find an increase of backup needs energy due to strong climate change in a wind-powered electricity system. This increase is solely rooted in changes of the wind resource since all other parameters are kept constant. In order to single out climate change induced alterations, we fix the technological parameters such as hub heights or turbine efficiencies, and we do not account for changes in the consumption such as load shifting or sector coupling throughout the 21st century.

For the identification of changes in the spatial wind patterns, we perform a correlation analysis over 20 year time-spans of wind speeds (see Table 1). We use Pearson correlation on the highest spatial scale, i.e. we correlate every grid point to all others instead of aggregating the wind fields first. Hence, the full spatial detail of the downscaled climate data is taken into consideration. In order to visualize results, correlation values are averaged on country level in the next step. To highlight long-term trends, we only show correlation changes between 2080-2099 (endc) and 1985-2004 (ref):

$$\Delta R_{\text{endc}}(\text{A,B}) = R_{\text{endc}}(\text{A,B}) - R_{\text{ref}}(\text{A,B}), \tag{9}$$

where $R_{\text{period}}(\text{A,B})$ denotes the average of all point-to-point correlations between country A and country B in a given period. The computation is repeated for all possible combinations $(A, B)$. We calculate $\Delta R_{\text{endc}}(\text{A,B})$ for each climate model separately and show the model mean if not stated otherwise.

To reveal general patterns, we first consider the average correlation change of a fixed country A by averaging Eq. (9) over all countries B excluding A (cf. Fig. 4). There is a general tendency towards higher correlations of wind speeds for Central Europe in the ensemble mean. This change is most pronounced in Germany, Switzerland, Benelux and Ireland. Decreasing correlations only occur at the fringes of the continent and they are strongest in Portugal and Greece. Positive correlation changes occur in most countries and the maximum positive change is approximately three times larger in magnitude than the maximum negative change. Interestingly, the overall pattern is similar to the mismatch contribution analysis (cf. Fig. 3). This similarity is not a trivial finding since the mismatch contribution analysis accounts for the non-linear turbine power curve and the collective behaviour of the entire electricity grid while the correlation analysis is solely based on wind speeds. Summarizing, we find more homogeneous wind conditions over most of the continent while the fringes decouple slightly. Results for mid century are weaker but clearly similar (cf. Supplementary Fig. 12).

Assessing pairwise correlation changes between countries, we find that the correlation increase over Central Europe has at least a high agreement in the EURO-CORDEX ensemble (cf. Fig. 5). Some country combinations (e.g. DE-CZ, FR-CZ, BE-GB, FR-NL) even show robust trends. For example, in Germany the correlations to all neighbour contries plus the British Isles and Eastern Europe increase with high agreement. The importance of this finding is strengthened by the fact that Central Europe plays an important role for the power system: Germany, France, Great Britain, Poland

and Benelux account for more than half of the European load. Correlations between Germany and Greece decrease with high model agreement. In contrast, changes between Germany and the Iberian Peninsula, Italy and Norway are uncertain.

The decoupling of Portugal and Greece which is found in the aggregated plot (Fig. 4) is only robust in a few country combinations and models disagree regarding some important pairs (e.g. PT-DE, PT-FR, PT-GB or GR-IT, GR-UK or ES-FR, ES-GE). The uncertainty with respect to the correlation changes between these countries is thus high.

However, a robust trend is found in Scandinavia, where Norway, Finland and Sweden become higher correlated. This change partly also holds for the Baltic region. At the same time Scandinavia decouples robustly from some parts of Southern Europe (e.g. SE-GR, NO-ES). In the context of large scale European grid expansions, these alterations might enhance the value of high-voltage direct current (HVDC) lines between these distinct regions.

Correlation increases in Scandinavia are also robust in the middle of the century (cf. Supplementary Fig. 13). However, inter-model agreement for correlation increases in Central Europe is lower albeit the overall pattern is still conceivable. The decoupling of Portugal and Greece can be seen in the inter-model mean while agreement across models is rare.

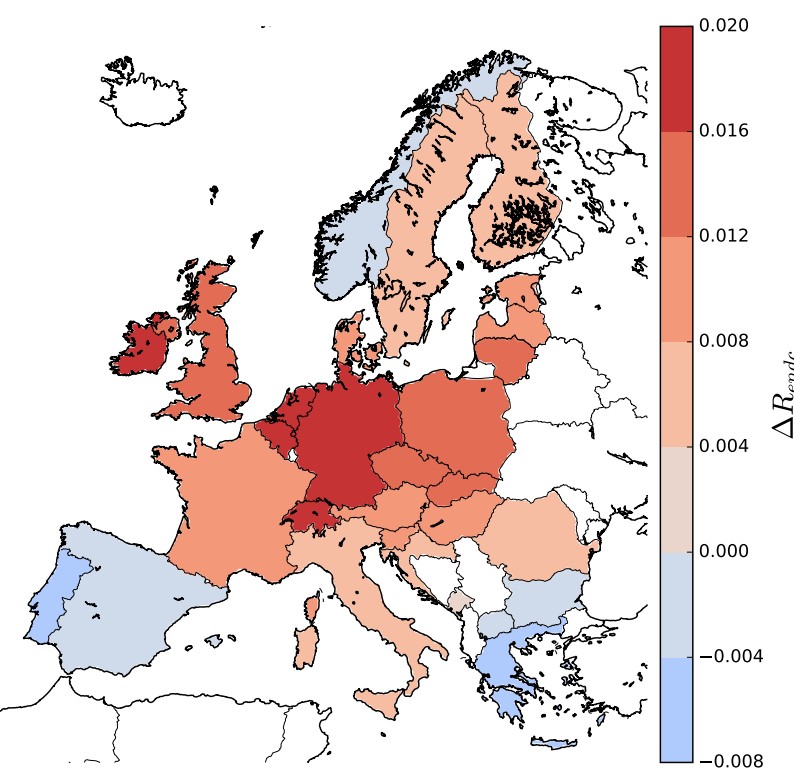

**Figure 4. Correlation changes of wind timeseries averaged over all models (difference between end of century and reference correlations).** An increase of spatial correlation over most of Europe is found which hints to more homogeneous wind conditions. This increase is most pronounced in the Central European region. However, at the margins of the continent correlation decreases are found. A more detailed assessment, which in particular addresses inter-model spread, is shown in Fig. 5.

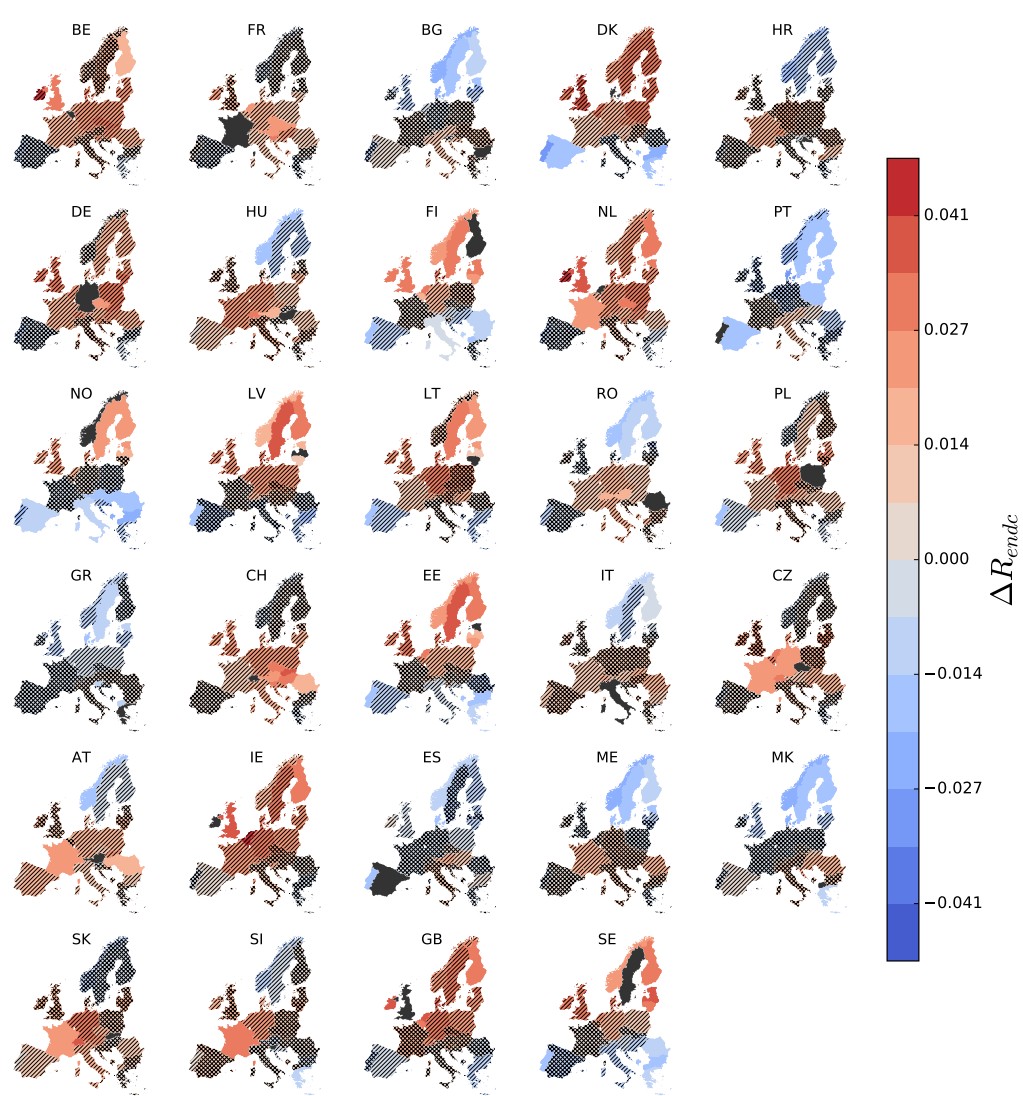

**Figure 5. Country-specific change of wind speed correlations at the end of the 21st century including inter-model agreement.** Colors denote the model-average correlation change of a country to the reference country (highlighted in black and given in the respective heading). Hatches indicate inter-model agreement as follows: no hatches indicate perfect agreement on sign of change, striped: 4 out of 5 models agree, crosses: less than 4 agree.

### 3.3 Climate: Complementing EURO-CORDEX with CMIP5 using Circulation Weather Types

The EURO-CORDEX dataset includes only a 5-member subset of all CMIP5 GCMs and might thus not be representative for the entire CMIP5 ensemble. Moreover, subgroups of GCMs can be biased in the same way since they did not develop separately, but along the same lines. The most drastic example is the sharing of code by CNRM and EC-EARTH, which are both part of the EURO-CORDEX ensemble and run the same atmosphere module (Knutti et al., 2013).

Uncertainty in climate projections has been argued to stem from three main sources: (1) natural variability, (2) model uncertainty and (3) scenario uncertainty (Hawkins and Sutton, 2009). In some situations the choice of initial conditions also contributes substantially (Hawkins et al., 2016). We neglect scenario uncertainty by design of this study since we only focus on the sensitivity to strong climate change (RCP8.5). As the importance of natural variability decreases with the time intervals averaged over, model uncertainty is likely to be the dominant source of uncertainty here.

In order to rule out the possibility that our findings are biased due to the (arbitrary) choice of GCMs that were downscaled for EURO-CORDEX, we follow a statistical-dynamical approach which was developed by Reyers et al. (2015, 2016) to downscale a large CMIP5 ensemble for wind energy applications. This approach is based on a circulation weather type (CWT) classification methodology (Jones et al., 1993). Daily mean sea level pressure (MSLP) values at 16 GCM grid points around a central point located in Germany are used to assign the near-surface atmospheric flow over Europe to either a directional flow (north, northeast, east, ...) and/or a rotational flow (anti-cyclonic, cyclonic). Aside from the direction of the atmospheric flow a $f$-parameter is calculated, which is representative for the instantaneous pressure gradient and thus for the general wind speed conditions over Germany and the surrounding countries.:

$$f = \sqrt{dP_z^2 + dP_m^2}, \tag{10}$$

where $dP_z$ is the mean pressure gradient in East-West direction (zonal component) and $dP_m$ is the mean pressure gradient in North-South direction (meridional component). $f$-parameters from below 5 hPa per 1000 km (weak MSLP gradient and thus low wind speed conditions) up to 45 hPa per 1000 km (strong MSLP gradients and thus high wind speed conditions) were found. Reyers et al. (2016) demonstrated that such a CWT classification provides a suitable and effective basis for wind energy applications on the regional scale and therefore enables the consideration of a large CMIP5 ensemble in future projections.

Analyzing the five individual GCMs contributing to the EURO-CORDEX ensemble reveals a link between the CWTs and the backup energy needs derived from dynamically downscaled data (see Eqs. 1, 2, 3). We find that backup needs energy decreases monotonously with increasing $f$-parameter (cf. Fig. 6a,b). All models in the EURO-CORDEX ensemble agree on this result which is also physically plausible as the pressure gradient drives the atmospheric circulation. This statement holds for Germany and its neighbors and for Europe as a whole. We see this as evidence that the

CWT analyses in this particular case can be applied to the entire continent in the sense that the $f$-parameter is a reasonable proxy for the European backup ~~need~~ energy.

The majority of CMIP5 models (16 out of 22) predicts an increase of events with low $f$-parameter by the end of the century (cf. Fig. 6c). Following the likelihood classification developed for the latest IPCC report (Mastrandrea et al., 2010), it is thus *likely* that low $f$-parameters become more abundant. This trend originates mainly from more frequent anticyclonic pressure configurations (cf. Fig. 7). For this CWT, spatial homogeneity of the wind resource is higher as compared to all other CWTs

(cf. Supplementary Fig. 14). In such a homogeneous situation, it is plausible that backup ~~needs are~~ energy is elevated since countries are more likely to experience shortfall of generation simultaneously. In contrast, medium ($10 \leq f[hPa/1000km] \leq 15$) and high ($15 \leq f[hPa/1000km] \leq 20$) $f$-parameters are *likely* to occur less frequent since 17 models agree on these signals. We thus conclude that the majority of CMIP5 models agrees with the main finding of increasing backup ~~needs~~

energy .

The larger CMIP5 ensemble also allows for an assessment of the EURO-CORDEX ensemble's input data. We report that the GCMs contributing to EURO-CORDEX are within the spread of the remaining CMIP5 ensemble (exception HadGEM for very strong f-parameters) and are thus generally representative for the larger ensemble (see Fig. 6). However, they also show comparably

strong changes in the occurence of specific $f$-parameters. The CMIP5 overall projection regarding backup ~~needs~~ energy might thus be lower than results reported in this paper. In order to test this speculative hypothesis, a consistent downscaling of all CMIP5 models would be necessary, which is far beyond the scope of this article but should be tackled in future works.

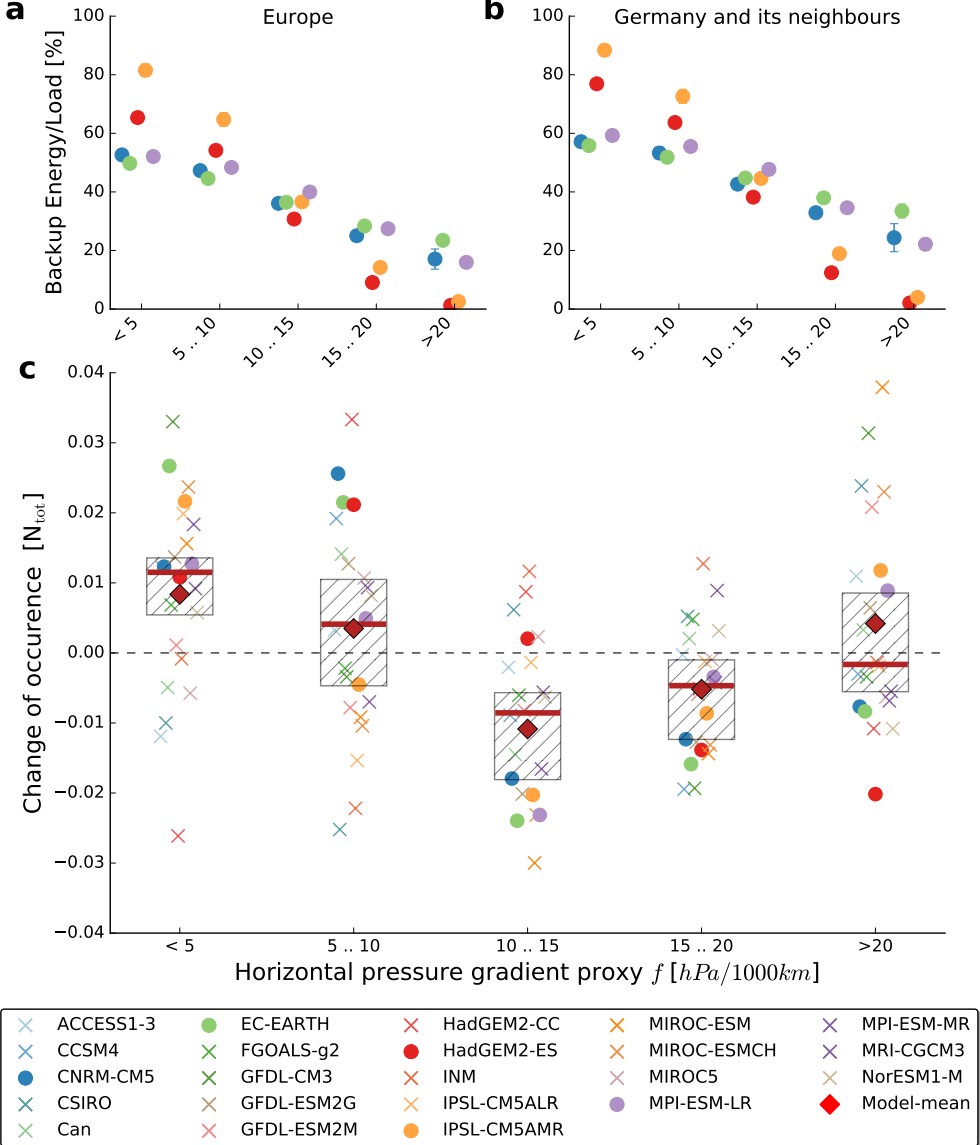

**Figure 6. Backup energy and change of occurrence as a function of the $f$-parameter. a**) Monotonous decrease of backup energy with increasing $f$-parameter Backup energy versus $f$-parameter for the entire domain. Circles denote the mean over the three considered periods for each model and errorbars indicate the standard deviation thereof. Errorbars are, however, most often smaller than the circle size. **b**) The same decline is found if only Germany plus its neighbor countries are considered. Same as **a**) but restricted to Germany and its neighbors **c**) Change of occurrence of different $f$-parameters. The change of occurence is computed as the difference between end of century and the reference period and is given in units of the total number of timesteps $N_{tot}$. Low $f$-parameters become more frequent by the end of the century while medium to high $f$-parameters occur less often. There is considerable inter-model spread, however 16/22 agree on an increase in frequency of very low pressure events ($f < 5$) and 17/22 agree on a decrease of medium pressure events ($10 \leq f < 15$). Red diamonds denote the ensemble mean, red lines the ensemble median and hatched boxes indicate the 33rd to 67th percentile. If a box lies completely above/below zero, the sign of the change can be considered as likely (Mastrandrea et al., 2010).

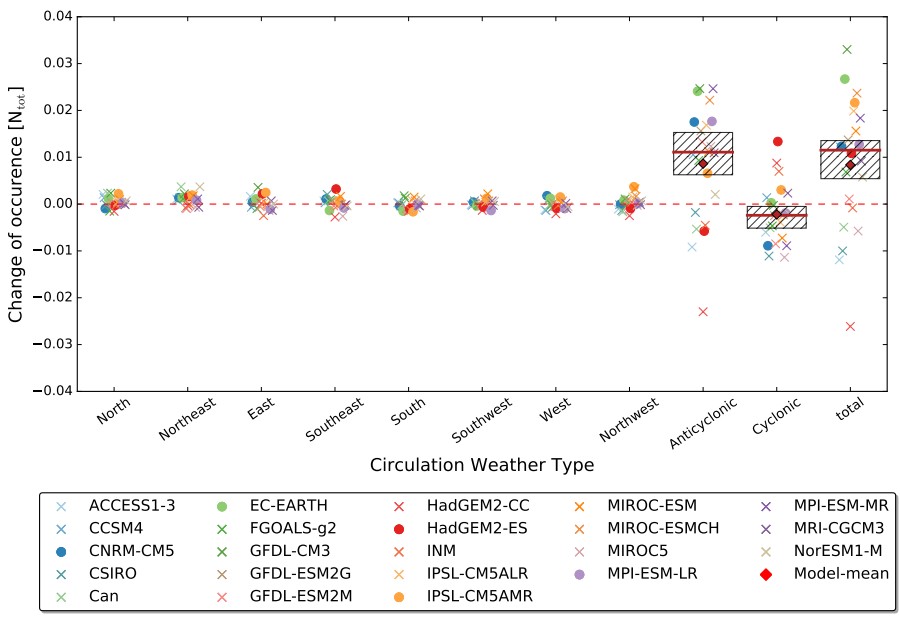

**Figure 7. Changes of relative occurrence of primary CWTs with low $f$-parameters ($f \leq 5hPa/1000km$).**
Changes are differences in occurrence between end of century and the reference period and are given in units of the total number of timesteps $N_{\text{tot}}$. Boxes indicate the 33rd to 67th percentile and are only shown if changes are substantial. A majority of models projects more weak anticyclones while cyclonic CWTs occur less often (both findings are likely). In total, most models project an overall increase of the occurence of CWTs with low $f$-parameters. This increase is dominantly rooted in more frequent anticyclonic CWTs. Red diamonds denote the ensemble mean and red lines denote the ensemble median.

## 4  Conclusions

A future highly-renewable electricity system will be governed by weather conditions. If mankind fails to reduce carbon emissions fast, climate change will impede the operation of a wind-driven system in Europe. This conclusion is based on three separate lines of evidence.

1. A coarse-scale electricity model fed with EURO-CORDEX climate data shows robust increases of backup needs energy.

2. Spatial correlations in wind timeseries in EURO-CORDEX data across Central Europe are found to increase. Countries are thus more likely to experience generation shortfall simultaneously.

3. Building upon a statistical-dynamical downscaling technique and a 22-member CMIP5 ensemble we find a likely increase of Circulation Weather Types with low $f$-parameters. They
are associated with low Europe-wide wind generation.

It has to be stressed that results are for the end of the 21st century and based on a strong climate change scenario (RCP8.5). They should be thought of as a sensitivity test. Moreover, While the increases of backup energy are robust yet , they are also restricted to relative increases of 87% (cf. Fig. 2). A fully-renewable electricity system will hence not become unfeasible due to catastrophic
changes.

In the emerging field of linking energy and climate research, many additional questions are to be addressed in order to deliver a more holistic assessment. We simulated a wind-driven electricity system and performed a control simulation with a fixed share of PV. Timeseries for the latter were taken from a validated dataset based on reanalysis data (Pfenninger and Staffell, 2016). Ideally, future
works would assess the combined effects of climate change on wind and solar generation. They could also include concentrated solar power since this technology bears advantages for system integration (Pfenninger et al., 2014). Load-shifting, sector-coupling and storage are further key words for more detailed assessments.

In terms of climate modeling output, a larger high-resolution ensemble is desirable which in par-
ticular contains multiple Regional Climate Models (RCMs). The next generation of CORDEX is planned to deliver such data (Gutowski Jr. et al., 2016) and will hence allow for an inclusion of RCM-spread in future assessments. It will also facilitate similar assessment for other world regions as spatial extent will be expanded.

**Contributions**

Ja. W. performed the simulations, analyzed the data, produced all figures and wrote most of the manuscript. D. W. conceived and supervised the research and contributed to the writing, in particular regarding the electricity system. M.R. supplied the CWT analysis and wrote parts of the CWT chapter. All authors contributed ideas, gave feedback and helped to improve the manuscript.

**Acknowledgements**

We thank M. Greiner, G. Andresen, S. Kozarcanin, T. Brown, D. Schlachtberger and C. Ball for stimulating discussions. We owe J. Brugger gratitude for checking the final manuscript. We also acknowledge the World Climate Research Programme's Working Group on Regional Climate, and the Working Group on Coupled Modelling, former coordinating body of CORDEX and responsible panel for CMIP5. We thank the climate modelling groups for producing and making available their model output. We acknowledge the Earth System Grid Federation infrastructure an international effort led by the U.S. Department of Energy's Program for Climate Model Diagnosis and Intercomparison, the European Network for Earth System Modelling and other partners in the Global Organisation for Earth System Science Portals (GO-ESSP). The authors gratefully acknowledge the computing time granted by the JARA-HPC Vergabegremium on the supercomputer JURECA (Juelich Supercomputing Centre, 2016) at Forschungszentrum Jülich. We gratefully acknowledge support from the Helmholtz Association (via the joint initiative "Energy System 2050 - A Contribution of the Research Field Energy" and the grant no. VH-NG-1025 to D. W.).

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
