# Peer review of "More Homogeneous Wind Conditions Under Strong Climate Change Decrease the Potential for Inter-State Balancing of Electricity in Europe"

_Earth System Dynamics, 2017_

## Referee Comment (RC2)

**Review: esd-2017-48 - More Homogeneous Wind Conditions Under Strong Climate Change Decrease the Potential for Inter-State Balancing of Electricity in Europe**

Jan Wohland, Mark Reyers, Juliane Weber, and Dirk Witthaut

August 18, 2017

**Comments to the Authors**

The manuscript under review presents a study of the impacts of strong climate change on the operation of a fully renewable European power system using future projections from the EURO-CORDEX dynamically downscaled regional climate ensemble. To assess the impact the authors compare historical and future backup energy needs, allowing for trans-national transmission. The authors consider this 'network expansion' as the cost-optimal adaptation strategy, rather than exploring the incorporation of storage capacity.

**Major Points**

I consider that this manuscript should be subject to minor revision due to the fact that the analysis of the results if often unclear given their definitions and use for expressions such as 'backup energy' and 'backup needs'. Given that the article has been submitted to a journal where authors and readers come from a diverse range of backgrounds, I believe that a clear nomenclature is fundamental.

Instances of these conflicts, along with an extended set of minor points is included next, with suggestions on how to improve the manuscript.

**Minor Points**

1. Page 3, lines 60-61: extra parenthesis in citation

2. Page 3, line 32: high resolution future projections but coarse representation of the power system? This might require a better description of the implications and assumptions. Are you considering the effect of changes in the national grids negligible?

3. Page 5, lines 92-94: This paragraph looks out of place in the Methods section and is redundant to the Introduction

4. Page 5, line 102: should be 'sensitivity analyses' or 'a sensitivity analysis'

5. Page 6, eqs 1-2: You don't include any representation of existing storage capacity in the system? How would results change if you did?

6. Page 6, line 51: a 20yr time slice only allows to account for a portion of natural variability: interannual rather than decadal, and you mention in your introduction that larger time-scales also have an impact on the power system operation

7. Figure 1 and Table 1 captions: These are very long. Consider including more of this information (which even includes multiple references!) in the Methods section.

8. Figure 2: one really can't tell much from panel a on this figure. Consider removing it and using only the changes, or just show it for fewer expansion coefficients (or just no expansion) to be able to 'zoom in'. On the caption, there is a typo and should read 'later on'. And you are also discussing a lot of the results on the caption!

9. Page 10, lines 75-77: The assessment is not clear. An increase in backup energy needs implies under your definition that there is more of the local energy mismatch (difference between demand and volatile RE) that could not be met by transmissions. So, how can this be due to more excess energy? Is the problem on the assessment of line 74, since the increase is not on backup NEEDS but rather on energy available for transmission? If what you show on the plot (panel a) is back up energy, that is decreasing with network expansion. You can see how your descriptions are leaving big gaps in the interpretation of results.

10. Page 10, lines 183-184: this is not true for all ensemble members. Cahnges in CNRM and MOHC are not that 'pronounced'.

11. Figure 8b, Supplementary Information: How can the backup energy increase by incorporating PV? You can see it in the two larger $\alpha$ values for the CNRM model in the *midc* period.

12. Page 10, line 194: should be 'reveal'

13. Page 11, lines 199-201: how is this a lower bound to 'back up needs' if this is this represents the worst case scenario for mismatch. I can see how it is a lower bound for the mismatch $M_i$, since it is negative.

14. Page 12, line 232: should be 'importing'

15. Page 13, lines 257-259: this clarification should have been made in the Methods section, since it was also an assumption of the previous analysis.

16. Figure 7: check language of labels in x axis. To mix the directions and rotations in the same plot makes it impossible to see any changes in the first. Consider adding two panels!

17. Page 20, lines 345-6: It seems like you are making assumptions about more spatially homogeneous condition from an analysis that is based on a single point. How can you draw those conclusions from CWT?

18. Page 23, lines 372-2: Comment starting in 'Moreover...' need revision

---

## Referee Comment (RC1) · Anonymous Referee #1 · 5 Jul 2017

Review of "More homogeneous wind conditions under strong climate change decrease the potential for inter-state balancing of electricity in Europe" by Wohland, Reyers, Weber, and Witthaut, submitted to Earth System Dynamics

The manuscript is very good and it is so well written and logically presented that it was a pleasure to review. I only have a few minor points.

Minor points

1) The captions of tables and figures are very long and include details on methods and even on findings. I found it useful actually, but captions should describe the figure or table, nothing more and nothing less. Maybe consult with the journal editors for guidance. You can easily move some of the caption text into the main text.
2) The abstract tends to over-emphasize the results without actual quantifications, which could be misinterpreted and used against wind energy if taken out of context. For example, rephrase as this: "… we find a robust but modest increase (up to 7%) of backup needs…" and "… resulting in parallel generation shortfalls of up to XX MW (corresponding to YY% of power demand) in up to ZZ% of the countries."
3) Line 110: please explain how the extrapolation to 80 m was done. Log law? Power law? Interpolation of model levels?
4) Line 111: which standard power curve was used? How were wake losses accounted for?
5) Line 113: Why were the wind farms sized at 100 MW?
6) Table 1: this table is not needed and could easily be incorporated either in the main text or in the legend/caption of Figure 2.
7) Figure 2: please use the same colors for the 5 models as in Figure 6 and 7 for consistency.
8) Figure 2: What are the units of a) and b)? $L_{ref}$? Shouldn't it be percent?
9) Figure 2c: Do you really need this figure? It has the same pattern as b) and it is difficult to conceptualize/understand. Also, having 2 figures instead of 3 would make them more readable. Right now they are too small.
10) Line 209: Is L the same as generation shortfall? Please mantion in what units it is expressed (MWh/yr)
11) Around line 215: Please compare the values of L with the total energy or capacity of each country. For example, from Figure 3 the maximum size of L is around 250 TWh/yr, which is possibly small for Germany but would be large for Hungary. Maybe a fraction of total electricity consumption should be used instead? Basically, we need a sense of how significant a given value of L is.
12) Line 324: please provide a definition/formula of f. Is it the Coriolis parameter?
13) Figure 6: cannot see the error bars in a) and b).

Typos/spelling

1) line 60: double parenthesis and note that you need a comma after e.g. "(e.g., Chiacchio et al. 2015; Herwehe et al. 2014)."

2) line 92: double parenthesis and note that you need a comma after e.g. "(e.g., Bloomfield et al. 2016)."

---

## Author Comment (AC1) · 31 Aug 2017

We thank both reviewers for their valuable feedback which helped to improve the manuscript substantially. We also thank the editor for handling our manuscript.

We provide our final author response and the new version of the manuscript high-lighting all changes attached to this comment. Please note that both documents are merged in one file because the web interface only allows to upload a single file as supplement.

Yours sincerely,

[Figure]

Jan Wohland (for all authors)

Please also note the supplement to this comment:
https://www.earth-syst-dynam-discuss.net/esd-2017-48/esd-2017-48-AC1-
supplement.pdf

[Figure]

**Supplement:**

**Final author comments: More Homogeneous Wind Conditions Under Strong Climate Change Decrease the Potential for Inter-State Balancing of Electricity in Europe**

**esd-2017-48**

Jan Wohland et al.

August 31, 2017

1 Throughout this manuscript red denotes deletions from the original manuscript and green
2 denotes additions to the text.

**Reviewer 1**

4 We thank the reviewer for his/her helpful comments to improve the manuscript. The points of
5 criticism were clearly formulated and mostly straightforward to implement. We are confident
6 that the modifications helped enhance the readability and avoid misinterpretations.

**Minor Point 1**

8 The captions of tables and figures are very long and include details on methods and even on
9 findings. I found it useful actually, but captions should describe the figure or table, nothing
10 more and nothing less. Maybe consult with the journal editors for guidance. You can easily
11 move some of the caption text into the main text.

**Author's response**

13 This aspect has been repeatedly mentioned by both reviewers (see Reviewer 2, Minor Points 7,8).
14 We hence decide to shorten captions where possible and move interpretations and discussion to
15 the main text.

**Changes in the manuscript**

**Caption of Fig. 1, p. 8:**

[revised manuscript text omitted]

**Minor Point 2**

The abstract tends to over-emphasize the results without actual quantifications, which could be misinterpreted and used against wind energy if taken out of context. For example, rephrase as this: ... we find a robust but modest increase (up to 7%) of backup needs... and ... resulting in parallel generation shortfalls of up to XX MW (corresponding to YY% of power demand) in up to ZZ% of the countries.

**Author's response**

We thank the reviewer for his/her comment to this very important section of the paper and propose to include his comments as given below.

**Changes in the manuscript**

Following a high emission pathway (RCP8.5), we find a robust but modest increase (up to 7%) of backup needs in Europe until the end of the 21st century. The absolute increase of the backup needs is almost independent of potential grid expansion, leading to the paradoxical effect that relative impacts of climate change increase in a highly interconnected European system. The increase is rooted in more homogeneous wind conditions over Europe resulting in extensive parallel intensified simultaneous generation shortfalls. Individual country contributions to European generation shortfall increase by up to 9 TWh/y, reflecting an increase of up to 4%. Our results are strengthened by comparison with a large CMIP5 ensemble using an approach based on Circulation Weather Types.

**Minor Point 3**

Line 110: please explain how the extrapolation to 80 m was done. Log law? Power law? Interpolation of model levels?

**Author's response**

When writing the manuscript, we decided to give this information in the supplement in an attempt to keep the manuscript concise. The Supplementary Material, A) Detailed Methodology states:

"Adopting the approach of Tobin et al. (2016), we use near-surface wind speeds 10 meters above the ground. Assuming a power-law relationship for the vertical wind profile, the velocity at hub height $H$ is obtained as

$$v_H = v_{10\mathrm{m}} \cdot \left(\frac{H}{10}\right)^{\frac{1}{7}} \tag{1}$$

and we chose $H = 80$m."

However, since there are multiple ways to perform the vertical scaling, we agree that it is important to modify the main text such that it includes the words 'power law'.

**Changes in the manuscript**

See Reviewer 1, Minor Point 4 as this deals with the same sentence.

**Minor Point 4**

Line 111: which standard power curve was used? How were wake losses accounted for?

**Author's response**

Similar to the previous point, we thought that the specifics of the calculation are ideally given in the Supplement. The Supplementary Material, A) Detailed Methodology states:

"The conversion of wind speeds into renewable generation is performed using a simple power curve

$$P(v_H) = P_0 \begin{cases} 0, & \text{if } v_H < v_i \text{ or } v_H > v_0 \\ \frac{v_H^3 - v_I^3}{v_R^3 - v_I^3}, & \text{if } v_I \leq v_H < v_R \\ 1, & \text{if } v_R \leq v_H < v_0 \end{cases} \tag{2}$$

where $v_H$ denotes wind velocity at hub height and $v_I = 3.5$ m/s, $v_R = 12$ m/s, $v_0 = 25$ m/s denote the cut-in, rated and cut-out velocity of the wind turbine, respectively."

Wake losses are not accounted for despite the fact that they reduce wind park yields. We argue that our findings are not severely impacted by this simplification. This is mainly because we focus on changes in wind generation and ignore wake losses both in the reference period and in future periods. While wake losses are likely to change absolute results, it seems plausible that they would impact system operation in the same way under current and future climate conditions. Moreover, it is -to our knowledge- still state-of-the-art either to neglect wake losses (e.g., Andresen et al., 2015) or to apply a bias-correction with measured generation data (e.g., Staffell and Pfenninger, 2016; Gonzalez Aparcio et al., 2016). In addition to wake losses, the bias-correction combines a large number of effects (e.g. unresolved orography, low temporal sampling, local wind phenomena, siting of wind parks, model errors). We are not aware of any reason that this aggregate of effects has to remain constant in time. For a long-term climate change study we thus argue that no bias-correction does less harm than a potentially wrong bias correction. However, it is desirable to develop a process-based representation of wake effects for future research. This would require to combine regional climate models and electricity system models rather than feeding the output of climate models into electricity models.

We agree with the reviewer that (a) information about the power curve should be already given in the methods section and (b) a clear statement that wake losses are neglected should be included.

**Changes in the manuscript**

Near-surface wind speeds are scaled up to hub height (80 m) based on a power law and a standard power curve is used to obtain the power generation of the wind turbines, both as in Tobin et al. (2016) (see also Supplementary Material A). The power curve assumes a cut-in velocity of 3.5 m/s, a rated velocity of 12 m/s and a cut-out velocity of 25 m/s. Wake losses are not accounted for. The country-wise aggregated ...

**Minor Point 5**

Line 113: Why were the wind farms sized at 100 MW?

**Author's response**

In principle, we follow the partially random allocation approach of Monforti et al. (2016). They argue that "the spatial allocation of future wind turbines (...) is difficult to forecast, as the localization process is dependent on social as well as economic and practical aspects, and are thus generally difficult to investigate." However, they find that "the actual deployment of national wind turbine fleets in 2020 in a country is expected to have a little overall influence on the main features of the national wind power profiles".

The approach necessitates to define the power of a wind park unit. While Monforti et al. (2016) used wind parks of 20 MW, we decided to use 100 MW. The main reason for this choice is that we consider a scenario where 100% of electricity is generated from wind turbines (on average), while Monforti et al. (2016) use the EU 2020 plans that lead to 10% generation from wind turbines. That is, we use five times larger parks to produce ten times the amount of electricity thus leading to more 100 MW wind parks in our assessment than 20 MW wind parks in their assessment. We are hence confident that errors arising from the discretization of wind parks in our assessment are smaller than in Monforti et al. (2016).

Moreover, turbine capacity has increased substantially while wind turbines matured (Wiser et al., 2016) and benefits arise from combining multiple turbines as maintenance and construction costs can be reduced. We hence assume that a future increase in wind park size is plausible.

**Changes in the manuscript**

The country-wise aggregated wind power is obtained by summing the generation of 100 MW wind parks until the system is fully-renewable on average. The wind park size was chosen as a compromise between increasing turbine capacities (Wiser et al., 2016) and the need for a sufficient amount of distinct parks. Wind parks are deployed semi-randomly

**Minor Point 6**

Table 1: this table is not needed and could easily be incorporated either in the main text or in the legend/caption of Figure 2.

**Author's response**

We agree with the reviewer that the information could be easily incorporated into the text or into a caption. Nevertheless, we prefer to keep the table because it is a lot easier to refer to a table than some section of the text. Note that the table is referenced three times in different sections of the paper.

**Minor Point 7**

Figure 2: please use the same colors for the 5 models as in Figure 6 and 7 for consistency.

**Author's response**

We thank the reviewer for pointing us to this shortcoming and adopt Figure 2 accordingly.

**Changes in the manuscript**

see Reviewer 1, Minor Point 9

**Minor Point 8**

Figure 2: What are the units of a) and b)? Lref ? Shouldnt it be percent?

**Author's response**

a) and b) are given in units of the total European load $D_{\text{tot}}$. Unfortunately, the axis label still referred to an older version of the manuscript. Instead of $L_{\text{tot}}$ it should read $D_{\text{tot}}$ which is defined as

$$D_{\text{tot}} = \int \sum_i D_i(t)dt. \tag{3}$$

Note that the demand is assumed to remain constant such that $D_{\text{tot}}$ is the same number for all periods.

We decided not to use percent as unit in a) and b) to facilitate understanding the difference to c). While a) and b) refer to absolute values (expressed as fractions of $D_{\text{tot}}$), c) refers to relative changes of the backup energy (both numerator and denominator depend on $E_B(ref)$).

If a) and b) were to be expressed in percent, the y-values would increase by a factor of 100.

**Changes in the manuscript**

The unit of the y axis in Fig. 2a,b is now $D_{\text{tot}}$ (see Reviewer 1, Minor Point 9). Moreover, the caption gives the definition ($D_{\text{tot}} = \int \sum_i D_i(t)dt$) of the total load (see Reviewer 1, Minor Point 1).

**Minor Point 9**

Figure 2c: Do you really need this figure? It has the same pattern as b) and it is difficult to conceptualize/understand. Also, having 2 figures instead of 3 would make them more readable. Right now they are too small.

**Author's response**

Both reviewers comment on this Figure (see Reviewer 2, Minor Point 8). Both propose to remove one panel in order to have more space for the individual subplots. Interestingly, Reviewer 1 suggests to remove c) stating that it has the same pattern as b) while Reviewer 2 suggests removing (or shrinking) a).

We think that the underlying problem is the size of the figure and we agree that it is too small. In terms of removing some content, we argue that all three panels add value and none of them can be easily left out because

a) allows for comparison of results with the literature (e.g., Rodriguez et al., 2014) and shows the potential range of backup energy reduction based on transmission. We suppose that this panel is particularly important for readers without a background in renewable energy integration as it gives an impression of scale and relevance: Roughly 45% of wind generation comes at the wrong time if no inter-country or temporal balancing is allowed.

b) indicates that the absolute increase is largely independent of grid expansion for three models. This is an indication for a large-scale effect and connects this section with the correlation and CWT analysis.

c) highlights that the relative change can be as high as 7 % and is substantially higher under strong grid extension. Given that current strategies of integrating renewables strongly build upon grid expansion, it is an important conclusion for policy making that those system are most vulnerable to climate change.

We thus suggest to keep all three panels but arrange them differently, such that readability is enhanced.

 **Changes in the manuscript**

[Figure]

Figure 1: **Updated version of Fig. 2**

**Minor Point 10**

Line 209: Is L the same as generation shortfall? Please mantion in what units it is expressed (MWh/yr)

**Author's response**

Yes, the unit of $L$ is $[L] = 1\frac{\text{TWh}}{\text{y}}$.

$L$ is not exactly identical to generation shortfall. It is the sum of local generation shortfalls *during European scarcity*. We propose to use both expressions ('energy that is lacking' and 'generation shortfall') because readers may disagree with respect to which one is more intuitive.

**Changes in the manuscript**

"We define the annual energy that is lacking (i.e., generation shortfall) in country i during European scarcity ... for convenience of interpretation. $L_i$ is given in TWh/y. A high value of ..."

**Minor Point 11**

Around line 215: Please compare the values of L with the total energy or capacity of each country. For example, from Figure 3 the maximum size of L is around 250 TWh/yr, which is possibly small for Germany but would be large for Hungary. Maybe a fraction of total electricity consumption should be used instead? Basically, we need a sense of how significant a given value of L is.

**Author's response**

We thank the reviewer for his/her comment which we have intensively discussed before submission. We think that the reviewer's comment raises the question of perspective. Different questions seem relevant or significant from different points of view. In Fig 3a we decided that we take a European perspective and ask: How much does every individual country contribute to the European problem (i.e. lacking energy $L_{\mathrm{ref}}$)? These numbers are biased in the sense that large consumers (such as Germany) have larger contributions than small consumers (like Hungary) due to their size.

One could certainly follow the reviewer's strategy and take a national perspective. What is the fraction of $L_{\mathrm{ref}}$ divided by the electricity consumption $D_i$ in each country? However, these numbers are also biased in another sense. If a big country has a small $L_{\mathrm{ref}}$ to $D_i$ ratio, it might seem to contribute little to the European problem even if it does contribute substantially.

Since both modes of presentation have their use and give answers to different questions, our idea was to show parts of both approaches. While Fig. 3a takes the European perspective, Fig. 3b takes the national one.

With respect to the units used, we argue that the choice is arbitrary and conventions seem to differ across disciplines. Our colleges dealing with energy system models prefer expressing energies in kWh and we decided to follow their convention.

In any case, we totally agree with the reviewer that the values of $L_{\mathrm{ref}}$ should be given for comparison. We thus provide the European aggregate value in the text and add a table to the Supplementaries giving the national values.

**Changes in the manuscript**

p. 12, line 213:

> ... whereas a low value of $L_i$ indicates a country whose generation shortfall can often be balanced by imports. In order to compare values of $L_i$ with loads, we provide country values for $D_i$ in the Supplementary Material E. The European sum is $\sum_i D_i \approx 3100$ TWh. Values for $\nu$ and $L$ during the reference period are shown in Fig. 3a,b. Large consumers like Germany and France are also the dominant contributors to European scarcity in terms of missing energy (cf. Fig. 3a). The German contribution corresponds to approximately 8% of the European annual load of 3100 TWh. However, the role of these countries, for example, in comparison to Eastern Europe or Benelux, is less pronounced if only the occurrence of negative mismatch events $\nu$ is considered...

We add the following table to the supplementaries:

Table 1: Annual sums of country electricity consumption based on hourly 2015 data provided by the European Network of Transmission System Operators for Electricity (2015).

| country | country code | Annual load [TWh] |
|---|---|---|
| Austria | AT | 69.62 |
| Belgium | BE | 85.22 |
| Bulgaria | BG | 38.62 |
| Switzerland | CH | 62.06 |
| Czech Republic | CZ | 63.53 |
| Germany | DE | 505.27 |
| Denmark | DK | 33.9 |
| Estonia | EE | 7.93 |
| Spain | ES | 248.5 |
| Finland | FI | 82.5 |
| France | FR | 471.26 |
| Great Britain | GB | 282.19 |
| Greece | GR | 51.4 |
| Croatia | HR | 17.19 |
| Hungary | HU | 40.75 |
| Ireland | IE | 26.57 |
| Italy | IT | 314.35 |
| Lithuania | LT | 10.86 |
| Latvia | LV | 7.07 |
| Montenegro | ME | 3.42 |
| Macedonia | MK | 7.84 |
| Netherlands | NL | 113.25 |
| Norway | NO | 128.65 |
| Poland | PL | 149.96 |
| Portugal | PT | 48.93 |
| Romania | RO | 52.31 |
| Sweden | SE | 135.93 |
| Slovenia | SI | 13.65 |
| Slovakia | SK | 28.21 |
| Total | | 3100.94 |

**Minor Point 12**

Line 324: please provide a definition/formula of f. Is it the Coriolis parameter?

**Author's response**

For the determination of the CWTs the sea level pressure at 16 horizontal grid points around a pre-defined central point (in this case near Frankfurt, Germany) is considered (see also Fig. 2 in Reyers et al., 2015). The f-parameter describes the mean horizontal pressure gradient over the domain defined by these 16 grid points and thus can serve as a measure for the wind speed conditions at the central point and the surrounding area.

$$f = \sqrt{dP_z^2 + dP_m^2} \tag{4}$$

where $dP_z$ is the mean pressure gradient in East-West direction (zonal component) and $dP_m$ is the mean pressure gradient in North-South direction (meridional component).

**Changes in the manuscript**

Aside from the direction of the atmospheric flow a $f$-parameter is calculated, which is representative for the instantaneous pressure gradient and thus for the general wind speed conditions over Germany and the surrounding countries.:

$$f = \sqrt{dP_z^2 + dP_m^2}, \tag{5}$$

where $dP_z$ is the mean pressure gradient in East-West direction (zonal component) and $dP_m$ is the mean pressure gradient in North-South direction (meridional component). $f$-parameters from below 5 hPa per 1000 km (weak MSLP gradient and thus low wind speed conditions)

**Minor Point 13**

Figure 6: cannot see the error bars in a) and b).

**Author's response**

This is probably because the error bars are most often smaller than the circles. However, they should be clearly visible for CNRM-CM (blue circles) and $f > 20hPa/1000km$. We agree that this is potentially misleading and hence propose to adapt the caption.

**Changes in the manuscript**

Circles denote the mean over the three considered periods for each model and errorbars indicate the standard deviation thereof. Errorbars are, however, most often smaller than the circle size.

**Spelling Comment 1**

line 60: double parenthesis and note that you need a comma after e.g. (e.g., Chiacchio et al. 2015; Herwehe et al. 2014).

**Author's response**

We correct the quotation accordingly.

**Spelling Comment 2**

line 92: double parenthesis and note that you need a comma after e.g. (e.g., Bloomfield et al. 2016).

**Author's response**

Quotation does not exist any more at this place. See Reviewer 2, Minor Comment 3.

**Reviewer 2**

We thank the reviewer for his or her clear comments and suggestions to improve the manuscript.

**Major Point 1/ General Comment**

I consider that this manuscript should be subject to minor revision due to the fact that the analysis of the results if often unclear given their definitions and use for expressions such as backup energy and backup needs. Given that the article has been submitted to a journal where authors and readers come from a diverse range of backgrounds, I believe that a clear nomenclature is fundamental. Instances of these conflicts, along with an extended set of minor points is included next, with suggestions on how to improve the manuscript.

**Author's response**

We fully agree that an interdisciplinary readership requires exact and clear language in order to enable everyone to follow the manuscript. With respect to the example of 'backup energy' and 'backup needs', we used them as synonyms because we thought some variety of language might make the reading more pleasant. However, we agree with the reviewer that this and other parts of the manuscript are confusing, and therefore decided to clarify it in the revised version.

**Changes to the manuscript**

1. We use the term 'backup energy' throughout the paper and substitute 'backup needs' with 'backup energy' in lines 5, 6, 123, 127, 131, 163, 171, 174, 178, 184, 201, 244, 248, 255, 334, 339, 345, 319, 355, 363,372.

2. We replace 'backup energy needs' by 'backup energies' in lines 150 and 333.

3. We give more details regarding the meaning of a coarse-scale representation of the power system (see Reviewer 2, Minor Point 2)

4. The derivation of the model equations is expanded to make it easier to follow for people without a background in energy related research (see Reviewer 2, Minor Point 9).

**Minor Point 1**

Page 3, lines 60-61: extra parenthesis in citation

**Author's response**

We corrected this mistake, see Spelling Comment 1 of Reviewer 1.

**Minor Point 2**

Page 3, line 32: high resolution future projections but coarse representa- tion of the power system? This might require a better description of the implications and assumptions. Are you considering the effect of changes in the national grids negligible?

**Author's response**

We assume that the reviewer refers to page 3, line 82.

The meaning of *'coarse scale view on the power system'* is that we neglect many details of real power systems (because they do not matter for the large-scale energy balance of generation and demand). For example, we neglect stability issues (e.g., n-1 criterion, supply of apparent power, cascading effects etc.). The real transmission network is furthermore a system of systems with different voltage levels designed to serve different purposes (transmission over long distances vs. appropriateness for end users) and it is controlled by different actors on multiple levels (e.g. Transmission System Operators and Distribution System Operators). This list is by no means complete and we do not try to capture any of these. Instead, what matters for changes in wind energy (balancing) potentials is a high-resolution representation of wind speeds both in space and time. Therefore we need *'high-resolution regional climate modeling results'*.

With respect to the reviewer's last question: On the contrary, we assume that all national grids have unlimited transmission capacities as explained in p.6 lines 137f. ('We assume all countries to run a loss-free and unlimited transmission network within their boarders.') and in lines 113f. ('The country-wise aggregated wind power is obtained by summing the generation of 100 MW wind parks until the system is fully-renewable on average.') That is, all countries expand their grids such that the maximum benefit from spatial balancing within the country is achieved. The approach is well established as similar representations of the power system have been employed in various earlier studies (Rodriguez et al., 2014, 2015b,a; Becker et al., 2014a,b; Schlachtberger et al., 2017).

**Changes in the manuscript**

In order to give an idea of assumptions behind our coarse scale approach and facilitate readability for an interdisciplinary audience we propose to add a sentence:

"In this article we study the impact of climate change on the operation conditions for future fully-renewable power systems. We combine the analysis and simulation of power systems with high-resolution regional climate modeling results to quantify changes in wind power generation. We adopt a coarse scale view on the power system to uncover the large-scale impacts of climate change. The coarse scale perspective neglects details that are irrelevant for the balancing of demand with wind generation such as supply of apparent power or different voltage levels in the grid. The focus of this study is to In particular, we address the potential of trans-national power transmission to cover regional balancing needs."

**Minor Point 3**

Page 5, lines 92-94: This paragraph looks out of place in the Methods section and is redundant to the Introduction

**Author's response**

We agree with the reviewer and delete the paragraph. The citation is added in the Introductory as the paper contributed substantially to the research field.

**Changes in the manuscript**

Page 5, lines 92-94

The power generated by wind turbines and solar photovoltaics is determined by the weather such that its variability crucially depends on atmospheric conditions (see, e.g. Bloomfield et al. (2016)). How does climate change affect these conditions and the challenges of system integration?

Page 3, lines 73-74

It is thus necessary to consider indicators such as the variability and synchronicity of generation in addition to total energy yields (Monforti et al., 2016; Bruckner et al., 2014; Bloomfield et al., 2016).

 **Minor Point 4**

 Page 5, line 102: should be sensitivity analyses or a sensitivity analysis

 **Changes in the manuscript**

 In the spirit of a sensitivity analyses analysis, we evaluate the representative concentration
 pathway RCP8.5.

**Minor Point 5**

Page 6, eqs 1-2: You dont include any representation of existing storage capacity in the system? How would results change if you did?

**Author's response**

Yes, we neglect storage in this paper. This is done on purpose following a separation approach. The issue of variable renewable generation can in principle be solved via (a) spatial balancing or (b) temporal balancing or any combination of them. The paper under review here follows strategy (a) while another paper from our group follows strategy (b) (Weber et al., 2017). We plan to combine both approaches in future work. However, in order to understand the coupled system, it is helpful to have understood the isolated systems first.

One main challenge in combining both strategies is to incorporate the decision making process. Assume a country which has sufficient renewable generation at a certain point in time to meet its own demand completely while its storage is half full. Would it aim to import electricity to further fill its storage? Or would it rather sell the energy it has stored? Or would it prefer not to do anything? This decision would also very likely depend on the forecasted generation for the next days. If lots of wind generation for the days ahead is predicted, the country would be more likely to sell its stored electricity. In restricting our analysis to one option at a time, these problems are muted for the moment. However, they do have to be tackled in future works.

Inclusion of the *current* storage capacities would have a small effect on backup energies. For example, the current German storage capacity is around $S = 0.04$ TWh (Weitemeyer et al., 2015) while the German annual electricity consumption is at the order of $D_{\mathrm{Ger}} = 500$ TWh. The fraction $S/D_{\mathrm{Ger}} = 8 \cdot 10^{-5}$ is hence small and allows to store a bit less than 45 minutes of average German load. Weitemeyer et al. (2015) study the effect of storage on a renewable German power system with a mix of PV and wind for different renewable penetrations. They find that a storage of 0.1 TWh would allow to reduce backup energy by around 5% as compared to a no storage scenario in a fully-renewable system (see Fig. 2 in Weitemeyer et al., 2015). The effect of *current* storage capacity on the system studied in our paper is considerably smaller because (a) the current storage size is only 40% of 0.1 TWh and (b) they incorporate 40% PV generation which can be more easily stored than wind because it dominantly follows a diurnal cycle. Potential backup energy reductions due to current storage are thus at the order of 1%. This is clearly smaller than the potential reductions from grid expansion studied here (roughly 15% for $\alpha = 10$, see Fig. 2a).

However, including *very large* storage infrastructure would even have the ability to reduce backup energies to zero in the simplified system studied here (if energy losses from conversion are neglected). This is due to to the long-term balance between generation and load (Supplementary, Eq. 4). Unlimited storage would shift excess energy from periods of overgeneration to periods when generation shortfall is experienced.

**Minor Point 6**

Page 6, line 151: a 20yr time slice only allows to account for a portion of natural variability: interannual rather than decadal, and you mention in your introduction that larger time-scales also have an impact on the power system operation

**Author's response**

We do agree that the sentence overstates and needs to be relativized since we certainly ignore variability on very long timescales. We also agree with the reviewer that a 20 year time slice does not allow to assess decadal variability in a meaningful way. However, we do not consider one 20 year time slice but five of them because we use the output of five different models. Since the models have no reason to be synchronized, it is plausible to assume that they are in different states with respect to modes of natural variability. A *robust* change across all models (such as the increase in backup energy reported in the paper) is hence likely not rooted in decadal variability with a recurrence time of a couple of decades.

**Changes in the manuscript**

We suggest to replace the sentence by the (slightly modified) more accurate explanation in the table caption

Time frames of 20 year duration are chosen to account for natural climatic variability (see Table 1).

Time frames are chosen to contain 20 years in order to capture natural variability of the climate system on a multi-year timescale while still ensuring that elapsed time between periods is long enough to consider them distinctly (see Table 1). Since GCMs do not reproduce natural variations synchronously (Farneti, 2017), robust signals found in the ensemble are very unlikely to be rooted in natural variations with a recurrence time of a couple of decades (such as the Atlantic Meridional Oscillation or the North Atlantic Oscillation; see Peings and Magnusdottir (2014) for a discussion of their role in mediating atmospheric conditions).

The new caption of table 1 then reads:

Periods are chosen to contain 20 years in order to capture natural variability of the climate system on a multi-year timescale while still ensuring that elapsed time between periods is long enough to consider them distinctly. Since GCMs do not reproduce natural variations synchronously (Farneti, 2017), robust signals found in the ensemble are very unlikely to be rooted in natural variations with a recurrence time of a couple of decades (such as the Atlantic Meridional Oscillation or the North Atlantic Oscillation; see Peings and Magnusdottir (2014) for a discussion of their role in mediating atmospheric conditions). The reference period ref ends before 2005 because GCMs in CMIP5 are driven by historic emissions only until this date and follow different representative concentration scenarios afterwards.

Periods used in this study. The reference period ref ends before 2005 because GCMs in CMIP5 are driven by historic emissions only until this date and follow different representative concentration scenarios afterwards.

**Minor Point 7**

Figure 1 and Table 1 captions: These are very long. Consider including more of this information (which even includes multiple references!) in the Methods section.

**Author's response**

We thank the reviewer for making us aware of this shortcoming.

**Changes in the manuscript**

We provide a common answer in Reviewer 1, Minor Point 1 as both critiques are identical.

**Minor Point 8**

Figure 2: one really cant tell much from panel a on this figure. Consider removing it and using only the changes, or just show it for fewer expansion coefficients (or just no expansion) to be able to zoom in. On the caption, there is a typo and should read later on. And you are also discussing a lot of the results on the caption!

**Author's response**

We thank the reviewer for his feedback on this Figure and agree largely.

**Changes in the manuscript**

The general criticism is similar to Reviewer 1, Minor Point 9 where we provide a common answer.

Moreover, we correct the typo and shorten the captions as given in Reviewer 1, Minor Point 1.

**Minor Point 9**

Page 10, lines 75-77: The assessment is not clear. An increase in backup energy needs implies under your definition that there is more of the local energy mismatch (difference between demand and volatile RE) that could not be met by transmissions. So, how can this be due to more excess energy? Is the problem on the assessment of line 74, since the increase is not on backup NEEDS but rather on energy available for transmission? If what you show on the plot (panel a) is back up energy, that is decreasing with network expansion. You can see how your descriptions are leaving big gaps in the interpretation of results.

**Author's response**

We suppose that there is a misunderstanding here which can easily be resolved. In the system under consideration, more backup energy directly leads to more excess energy and excess energy has to be curtailed. This is because we assume that renewables generate as much electricity as needed on average. The statement can be formally derived from Eq. (2) in the manuscript by summing over all countries $i$ and integrating over an entire period from $t_s$ to $t_e$ which yields:

$$\int_{t_s}^{t_e} \sum_i M_i(t)dt + \int_{t_s}^{t_e} \sum_i B_i(t)dt + \int_{t_s}^{t_e} \sum_i F_i(t)dt = \int_{t_s}^{t_e} \sum_i C_i(t)dt. \qquad (6)$$

Recall that $M_i$ is the mismatch, $B_i$ is backup power, $F_i$ denotes imports or exports and $C_i$ denotes curtailment. The first term in Eq. 6 vanishes because of the assumption of a fully-renewable system (cf. Supplementary Eq. 12). The third term also vanishes because every import in one country ($F_j > 0$) is an export in another ($F_k < 0$) such that in total all imports are balanced by exports ($\sum_i F_i(t) = 0$). It follows that

$$\int_{t_s}^{t_e} \sum_i B_i(t)dt = \int_{t_s}^{t_e} \sum_i C_i(t). \qquad (7)$$

The left hand side is the backup energy $E_\mathrm{B}$ as defined in Eq. (3) in the manuscript and the right hand side is European curtailment during a period. (Note that Eq. 3 in the manuscript includes a minimization of $B_i$ which is needed to determine the im- and exports. For an aggregated European assessment, the actual im- and exports do not matter since they cancel anyway as argued above.)

**Changes in the manuscript**

We suggest restructuring of the sentence as follows

> The increase implies more excess energy and also more curtailment since we consider a scenario where 100% of electricity is generated from renewables on average.

> Since we consider a scenario where 100% of electricity is generated from renewables on average, an increase of backup energy is accompanied by an increase of excess energy which has to be curtailed.

Moreover, we add some information to the methods section to enhance readability for non-experts in the field of energy research:

p. 6, lines 137-138: The assumption of a fully-renewable system means that all countries generate as much electricity as needed on average ($\int_{t_s}^{t_e} M_i(t)dt = 0$). Furthermore, we We assume all countries to run a loss-free and unlimited transmission network within their boarders.

p. 6, lines 139-147:

If a country has a negative mismatch ($M_i < 0$, red circles in Fig. 1d), it tries to import energy. If it has a positive mismatch ($M_i > 0$, green circles in Fig. 1d), it tries to export energy. For each country $i$ the power balance must be satisfied:

$$M_i(t) + B_i(t) + F_i(t) = C_i(t),.$$ (8)

The mismatch $M_i$ can be compensated either by power generation from conventional backup power plants ($B_i \geq 0$), the curtailment of renewable power generation ($C_i \geq 0$) or by imports ($F_i > 0$) or exports ($F_i < 0$). To utilize renewable generation in an optimal way, countries will first try to balance power using im- and exports. However, a perfect balancing of all nodes is impossible if there is a continent-wise shortage or overproduction. Furthermore, cross-boarder flows along lines are bound by the directional Net Transfer Capacities (NTCs; see Supplement A for details), which may also impede balancing for some nodes. Power balance must then be satisfied by local means: In the case of a shortage, power must be backed up by conventional generators ($B_i > 0$). where $F_i$ represents imports ($F_i > 0$) or exports ($F_i < 0$) to/from country $i$. Cross-boarder flows along lines are bound by the directional Net Transfer Capacities (NTCs; see Supplement A for details). If overall shortage or line limits prohibit sufficient imports, power can also be backed up locally ($B_i \geq 0$). Similarly, if excess power can not be exported, it has to be curtailed ($C_i \geq 0$). We recognize that the technical details of backup generation often matter for implementation (Schlachtberger et al., 2016) but we focus on gross electricity needs in this study.

Additionally we add on page 7, line 155:

The European amount of backup energy is identical to the amount of curtailment over a full period. This is a direct consequence of the assumptions made and can be formally derived by summing Eq. 6 over all countries and integrating over an entire period. Since $\int_{t_s}^{t_e} M_i(t)dt = 0$ (each country is fully renewable on average) and $\sum_i F_i = 0$ (all imports to one country $F_j = c$ are exports from another $F_k = -c$) it follows that:

$$\int_{t_s}^{t_e} \sum_i B_i(t)dt = \int_{t_s}^{t_e} \sum_i C_i(t).$$ (9)

A change of the backup energy thus directly implies a change in total curtailment.

**Minor Point 10**

Page 10, lines 183-184: this is not true for all ensemble members. Cahnges in CNRM and MOHC are not that pronounced.

**Author's response**

While there is also an increase for CNRM and MOHC, we agree that the sentence is not strictly true for the two models mentioned. Given that the subsequent sentence deals with the considerable inter-model spread we suppose to add another sentence after this one.

**Changes in the manuscript**

There is considerable inter-model spread regarding the magnitude of change which varies by up to one order of magnitude depending on the climate model (see Fig. 2b, $\alpha = \infty$). In particular, changes for CNRM are generally weak and HadGEM2 features only a slight overall increase with grid expansion.

**Minor Point 11**

Figure 8b, Supplementary Information: How can the backup energy increase by incorporating PV? You can see it in the two larger $\alpha$ values for the CNRM model in the *midc* period.

**Author's response**

If we understand correctly, the reviewer compares Fig. 2b in the manuscript and Fig. 8b in the supplement. While Fig. 2b always gives clearly negative absolute changes of the backup energy for CNRM, the change of backup energy approaches zero under $\alpha \in [10, \infty]$ in Fig. 8b. This means that the backup energy almost stays constant from *ref* to *midc* if PV is included, while it is reduced when PV is ignored.

To start with, backup energy (in absolute terms) is not higher but lower if PV is included as Figs. 2a and 8a show. For example, for $\alpha = 10$ the backup energy without PV is roughly $E_{\mathrm{B}} = 0.3 L_{\mathrm{tot}}$ while it comes down to roughly $E_{\mathrm{B}} = 0.25 L_{\mathrm{tot}}$ if PV is included. This decline is to be expected because wind and solar are to some extent complementary and their combination allows to reduce generation shortfall.

The observation that the backup energy decreases for small values of $\alpha$ in Fig. 8b indicates that the climatic changes are beneficial for the isolated or weakly connected European System following CNRM by midc. Grid expansion allows for spatial smoothing of the generation and brings backup energies down (cf. Fig. 8a). However, in a strongly connected system, no further positive effects due to climate change occur.

As a general note, PV is not considered additionally to wind generation in this study but rather as a *substitute* for a certain fraction of wind generation (cf. Supplementary ll. 607 ff.). In those scenarios where PV is included, only 71% of the load has to be met by wind (leading to fewer wind parks) while the remainder is provided by PV.

**Minor Point 12**

Page 10, line 194: should be reveal

**Changes in the manuscript**

Results are barely sensitive to changes in the load timeseries as an assessment using constant loads reveils reveals (cf. Supplementary C).

**Minor Point 13**

Page 11, lines 199-201: how is this a lower bound to back up needs if this is this represents the worst case scenario for mismatch. I can see how it is a lower bound for the mismatch Mi , since it is negative.

**Author's response**

We know that backup energy decreases monotonously with grid expansion (see Fig. 2a). This is because a well developed grid allows for spatial integration of volatile renewable generation. The case of unlimited transmission (i.e. $\alpha = \infty$) hence yields the lowest backup energies and provides a lower bound for backup energies. In other words, backup energies in a real system ($\alpha < \infty$) must be higher than the ones discussed in this section.

**Minor Point 14**

Page 12, line 232: should be importing

**Author's response**

The argument works in both directions. If a set of countries suffers from generation shortfall while Europe suffers from a generation shortfall, they can neither export electricity to alleviate the *overall* shortage nor import electricity to alleviate *their own* shortage. We prefer to keep the sentence as it is because we want to highlight that these countries can not contribute to solve the overall problem.

**Minor Point 15**

Page 13, lines 257-259: this clarification should have been made in the Methods section, since it was also an assumption of the previous analysis.

**Author's response**

We thank the reviewer and agree that the sentence fits better to the methods section.

**Changes in the manuscript**

We propose to include the sentence in the Methods section, p. 5, ll. 115

> ... following the approach of (Monforti et al., 2016). In order to single out climate change induced alterations, we fix the technological parameters such as hub heights or turbine efficiencies, and we do not account for changes in the consumption such as load shifting or sector coupling throughout the 21st century.

**Minor Point 16**

Figure 7: check language of labels in x axis. To mix the directions and rotations in the same plot makes it impossible to see any changes in the first. Consider adding two panels!

**Author's response**

We thank the reviewer for making us aware of the language issues and correct them accordingly.

We would like to stress that we consider 10 distinct CWTs. 8 of them are directional (N, NE, E etc.) and 2 of them are rotational (Anticyclonic, Cyclonic). There are no mixed CWTs in this assessment. The misunderstanding might originate from p.19 lines 321-324 where 'and/or' should read 'or'. We correct this mistake.

Based on this, it is interesting that most of the change is caused by rotational CWTs which is clearly visible in the plot as it is. If we were to use two different panels for directional and rotational CWTs, this information would potentially be masked. We therefore prefer not to add two panels.

**Changes in the manuscript**

[Figure]

Figure 2: **Updated version of Fig. 7.**

p.19 lines 321-324:

Daily mean sea level pressure (MSLP) values at 16 GCM grid points around a central point located in Germany are used to assign the near-surface atmospheric flow over Europe to either a directional flow (north, northeast, east, . . .) and/or a rotational flow (anticyclonic, cyclonic).

**Minor Point 17**

Page 20, lines 345-6: It seems like you are making assumptions about more spatially homogeneous condition from an analysis that is based on a single point. How can you draw those conclusions from CWT?

**Author's response**

As already stated in the submitted manuscript, for the determination of the CWTs the sea level pressure at 16 horizontal grid points around a pre-defined central point (in this case near Frankfurt, Germany) is considered (see also Fig. 2 in Reyers et al., 2015). Hence, the analysis is not based on a single point but on a horizontal pressure field covering large parts of the European sector. As a consequence, Reyers et al. (2015) could demonstrate that CWTs enable reliable conclusions about the regional wind conditions for a domain which covers Germany and the surrounding countries. It is thus possible to make assumptions about the spatial homogeneity, as stated in the submitted manuscript.

**Minor Point 18**

Page 23, lines 372-2: Comment starting in Moreover... need revision

**Author's comment**

We thank the reviewer for his comment and modify as given below. In particular, we correct the percentage from 8% to 7% which is more exact (see Fig. 2c) and in line with the number given in the abstract (see Reviewer 1, Minor Point 2).

**Changes in the manuscript**

Moreover, While the increases of backup energy are robust yet , they are also restricted to relative increases of 87% (cf. Fig. 2). A fully-renewable electricity system will hence not become unfeasible due to catastrophic changes.

**Other modifications**

**Update bibliography**

The paper (Schlachtberger et al., 2016) has been accepted in the meantime and is now referenced correctly.

**Substantial new work by Grams et al. (2017)**

Grams et al. (2017) showed that volatility of wind generation can be drastically reduced if wind park locations are chosen based on weather patterns rather than concentrated in the North Sea. We want to include the reference on page 13 line 242 as

> Moreover, Greece shows favourable changes for the European system in terms of energy contributions and occurrences with a high inter-model agreement (cf. Fig. 3c,d). This finding is particularly interesting as Grams et al. (2017) show that a combination of wind parks allocated in the North Sea and the Balkans allows to reduce volatility substantially under current climatic conditions. Based on our results, this positive effect from incorporating the Balkans would further be enhanced under strong climate change.

[revised manuscript text omitted]

**F) Annual load values on country level**

**Table 2.** Annual sums of country electricity consumption based on hourly 2015 data provided by the European Network of Transmission System Operators for Electricity (2015).

| country | country code | Annual load [TWh] |
|---|---|---|
| Austria | AT | 69.62 |
| Belgium | BE | 85.22 |
| Bulgaria | BG | 38.62 |
| Switzerland | CH | 62.06 |
| Czech Republic | CZ | 63.53 |
| Germany | DE | 505.27 |
| Denmark | DK | 33.9 |
| Estonia | EE | 7.93 |
| Spain | ES | 248.5 |
| Finland | FI | 82.5 |
| France | FR | 471.26 |
| Great Britain | GB | 282.19 |
| Greece | GR | 51.4 |
| Croatia | HR | 17.19 |
| Hungary | HU | 40.75 |
| Ireland | IE | 26.57 |
| Italy | IT | 314.35 |
| Lithuania | LT | 10.86 |
| Latvia | LV | 7.07 |
| Montenegro | ME | 3.42 |
| Macedonia | MK | 7.84 |
| Netherlands | NL | 113.25 |
| Norway | NO | 128.65 |
| Poland | PL | 149.96 |
| Portugal | PT | 48.93 |
| Romania | RO | 52.31 |
| Sweden | SE | 135.93 |
| Slovenia | SI | 13.65 |
| Slovakia | SK | 28.21 |
| Total | | 3100.94 |